# Single cell transcriptomics clarifies the basophil differentiation trajectory and identifies pre-basophils upstream of mature basophils

Kensuke Miyake [1,5] ✉, Junya Ito[1,5], Jun Nakabayashi[2], Shigeyuki Shichino [3], Kenji Ishiwata [4] & Hajime Karasuyama [1]

Basophils are the rarest granulocytes and are recognized as critical cells for type 2 immune responses. However, their differentiation pathway remains to be fully elucidated. Here, we assess the ontogenetic trajectory of basophils by single-cell RNA sequence analysis. Combined with flow cytometric and functional analyses, we identify c-Kit⁻CLEC12Aʰⁱ pre-basophils located downstream of pre-basophil and mast cell progenitors (pre-BMPs) and upstream of CLEC12Aˡᵒ mature basophils. The transcriptomic analysis predicts that the pre-basophil population includes previously-defined basophil progenitor (BaP)-like cells in terms of gene expression profile. Pre-basophils are highly proliferative and respond better to non-IgE stimuli but less to antigen plus IgE stimulation than do mature basophils. Although pre-basophils usually remain in the bone marrow, they emerge in helminth-infected tissues, probably through IL-3-mediated inhibition of their retention in the bone marrow. Thus, the present study identifies pre-basophils that bridge the gap between pre-BMPs and mature basophils during basophil ontogeny.

Basophils are the least common granulocytes, comprising <1% of circulating leukocytes. They have several similarities with tissue-resident mast cells, including the expression of the high-affinity IgE receptor (FcεRI) on their cell surface and the prompt release of granule contents (degranulation) upon cross-linking of FcεRI. Therefore, basophils had mistakenly been regarded as blood-circulating surrogates of mast cells. However, the development of analytical tools for basophils, such as basophil-deficient mice[1,2] and basophil-reporter mice[3,4], has enabled us to understand non-redundant functions of basophils in various immune responses[5–8], including chronic allergic inflammation, autoimmune diseases, and protective immunity against parasitic infections.

Basophils terminally differentiate within the bone marrow in contrast to mast cells, and several progenitor populations have been reported to differentiate into mature basophils[9]. Unipotent basophil progenitors (BaPs) are identified in 2005 and defined as Lineage (Lin)⁻cKit⁻CD34⁺FcεRIα⁺ cells[10] or Lin⁻cKit⁻CD34⁺CD200R3⁺ cells[11]. As BaPs can be generated in vitro from granulocyte-macrophage progenitors (GMPs), they are considered as down-stream of GMPs. Further studies identified bi-potential basophil and mast cell common progenitors in the bone marrow (pre-BMPs [Lin⁻Sca-1⁻cKit⁺CD34⁺FcεRIα⁺ FcγRII/IIIʰⁱ cells] and pro-BMPs [Lin⁻Sca-1⁻cKit⁺CD34⁺E-cadherin⁺FcγRII/IIIʰⁱ cells])[12,13]. In addition, bi-potential basophil and mast cell progenitor cells (BMCPs) have been identified also in the spleen[10], even

[1]Inflammation, Infection & Immunity Laboratory, Advanced Research Institute, Tokyo Medical and Dental University (TMDU), Tokyo, Japan. [2]College of Liberal Arts and Sciences, Tokyo Medical and Dental University (TMDU), Tokyo, Japan. [3]Division of Molecular Regulation of Inflammatory and Immune Diseases, Research Institute of Biomedical Sciences, Tokyo University of Science, Noda, Japan. [4]Department of Tropical Medicine, The Jikei University School of Medicine, Tokyo, Japan. [5]These authors contributed equally: Kensuke Miyake, Junya Ito. ✉e-mail: miyake.mbch@tmd.ac.jp

though later study has suggested that BMCPs can only differentiate into mast cells[14]. Thus, basophils differentiate from hematopoietic stem cells (HSCs) via GMPs, bi-potential basophil and mast cell progenitors and uni-potent basophil progenitors[9].

Recent advancements in the single-cell RNA sequencing (scRNA-seq) technology have challenged the classical view of hematopoietic cell differentiation, including the stepwise model of the hematopoietic differentiation tree. A series of single-cell transcriptomic analyses have provided a detailed snapshot of hematopoietic cell differentiation landscape, which indicates that hematopoietic cell differentiation is a rather continuous process in that oligo-potent progenitors differentiate into each respective cell lineage[15–17]. This was also the case in basophils, and dynamic gene expression changes from HSCs to basophils have been reported in mice and humans[18–22]. Findings from single-cell transcriptomes indicate that basophil differentiation trajectory is coupled with erythrocyte/megakaryocyte differentiation rather than neutrophil or monocyte differentiation. Indeed, several reports showed the presence of oligo-potent basophil and erythrocyte/megakaryocyte progenitors in mice and humans[18,23]. Thus, single-cell transcriptomes have shown differentiation pathways from HSCs to BaPs. Nevertheless, the continuous differentiation trajectory from progenitors to mature basophils remains to be fully defined, and it is uncertain whether intermediate populations exist between progenitors and mature basophils along the basophil ontogeny.

In this work, we define the basophil differentiation trajectory by conducting highly sensitive scRNA-seq analysis of basophils in mice and identify pre-basophils located between pre-BMPs and mature basophils during basophil ontogeny.

## Results

### Bone marrow-derived basophils (BMBAs) contain two subpopulations

Basophils are the rarest granulocytes with a short lifespan, and therefore BMBAs are commonly utilized as surrogates of basophils freshly isolated from animals[24]. To determine the culture condition optimal for the generation of BMBAs, we cultured mouse bone marrow cells with different concentrations of IL-3, ranging from 0 to 100 ng/mL, for 7 days. The frequency of CD200R3⁺c-Kit⁻ basophils in the culture increased up to ~60% as the IL-3 concentration went up to 0.3 ng/mL (Supplementary Figs. 1a, 2a, b). At higher IL-3 concentrations, the basophil frequency rather decreased, whereas the frequency of CD200R3⁺c-Kit⁺ mast cells increased (Supplementary Fig. 2a, b). Thus, 0.3 ng/mL of IL-3, much lower than commonly used concentration, turned out to be optimal for high purity of basophils.

In the above experiments, we noticed the presence of two subpopulations among the CD200R3⁺c-Kit⁻ BMBAs: FcεRIαhiCD49blo and FcεRIαloCD49bhi ones (Fig. 1a). The frequency of the former subpopulation decreased as the concentration of IL-3 increased regardless of mouse strains used, either C57BL/6 or BALB/c mice (Supplementary Fig. 2c–e), suggesting the influence of IL-3 on the balance between the two subpopulations. Notably, FcεRIαloCD49bhi BMBAs displayed ring-shaped nuclei as observed in peripheral blood basophils[25] whereas FcεRIαhiCD49blo BMBAs exhibited an atypical morphology with kidney-shaped indented nuclei and larger cell body (Fig. 1b). In accordance with their large cell body, FcεRIαhiCD49blo BMBAs displayed higher forward scatter (FSC) compared to FcεRIαloCD49bhi BMBAs in flow cytometric analysis (Fig. 1c).

In order to further characterize and compare these two subpopulations of BMBAs, we conducted unbiased single-cell RNA-seq (scRNA-seq) analysis of cells present in the IL-3 culture of bone marrow cells. Seurat clustering analysis of scRNA-seq data sets identified 11 clusters of cells (Supplementary Fig. 3a), in which clusters 0, 2, and 10 corresponded to basophils as judged by their high expression of *Cd200r3* and *Mcpt8* and little expression of *Kit* (Supplementary Fig. 3a, b). Among these three clusters (BMBA1, BMBA2, and BMBA3,

respectively, in Fig. 1d), BMBA2 and BMBA3 displayed lower *Fcer1a* (encoding FcεRIα) and higher *Itga2* (encoding CD49b) expression than did BMBA1 (Fig. 1d, e and Supplementary Fig. 3c). Therefore, BMBA2 and BMBA3 appeared to correspond to FcεRIαloCD49bhi BMBAs while BMBA1 corresponded to FcεRIαhiCD49blo BMBAs. Between BMBA1 and BMBA2/3, we detected 5382 differentially expressed genes (DEGs), in which 4811 genes were upregulated in BMBA1 while 571 genes were upregulated in BMBA2/3 (Fig. 1f). Among them, we picked up two genes coding for proteins expressed on the cell surface, *Clec12a* encoding CLEC12A and *Cd9* encoding CD9 (Fig. 1e, f), as possible surrogate markers of *Fcer1a* (encoding FcεRIα) and *Itga2* (encoding CD49b), respectively, for the use of discriminating two subpopulations of BMBAs in the following experiments. Indeed, the flow cytometric analysis using monoclonal antibodies specific to those proteins demonstrated that FcεRIαhiCD49blo and FcεRIαloCD49bhi BMBAs nearly correspond to CLEC12AhiCD9lo and CLEC12AloCD9hi BMBAs, respectively (Fig. 1g). Likewise, CLEC12AhiCD9lo and CLEC12AloCD9hi BMBAs nearly corresponded to FcεRIαhiCD49blo and FcεRIαloCD49bhi BMBAs, respectively (Fig. 1h).

### Two subpopulations of basophils are also detected in the bone marrow while only one of them in the peripheral blood and spleen

We next examined whether the two basophil subpopulations identified in BMBAs can also be detected in basophils freshly isolated from mice. Among c-Kit⁻CD200R3⁺CD49b⁺ basophils, the FcεRIαloCD49bhi/CLEC12AloCD9hi subpopulation was readily identified in the peripheral blood, spleen and bone marrow (Fig. 2a and Supplementary Fig. 1b). By contrast, the FcεRIαhiCD49blo / CLEC12AhiCD9lo subpopulation was rarely detected in the peripheral blood and spleen despite its presence in the bone marrow (Fig. 2a). CLEC12AloCD9hi basophils in the bone marrow, spleen and peripheral blood possessed typical ring-shaped or bi-lobulated nuclei while CLEC12AhiCD9lo basophils in the bone marrow displayed kidney-shaped indented nuclei with larger cell body and higher FSC in the flow cytometry than did CLEC12AloCD9hi basophils (Fig. 2b, c) as found in BMBAs. Of note, Lin⁻cKit⁻CD34⁺CD200R3⁺ BaPs also displayed the CLEC12AhiCD9lo phenotype (Supplementary Figs. 1c, 4a), and ~10% of the CLEC12Ahi population among cKit⁻CD200R3⁺ basophil-lineage cells showed low expression of CD34 (Fig. 2d and Supplementary Fig 4b), suggesting that CLEC12Ahi basophils might contain the uni-potent BaP-like population. Supporting this notion, CD34⁺ BaPs and CD34⁻ CLEC12Ahi basophils displayed similar morphology and surface expression profiles (Supplementary Fig 4c–e).

### scRNA-seq analysis identifies 4 clusters among bone marrow basophils

We sought to analyze the differentiation trajectory of bone marrow basophils by using *Mcpt8*GFP transgenic mice, where the GFP-encoding gene is expressed under the *Mcpt8* promoter/enhancer[4], for efficient enrichment and isolation of rare basophils. We confirmed that the majority of both CLEC12Ahi and CLEC12Alo basophil subpopulations expressed GFP (Supplementary Fig. 5a). Lineage (Lin)⁻GFP⁺ cells isolated from the bone marrow and spleen were separately subjected to scRNA-seq analysis (Supplementary Fig. 5b). Clustering analysis of scRNA-seq data identified 12 clusters, among which the clusters 0, 1, and 4 corresponded to *Kit*⁻*Cd200r3*⁺*Mcpt8*⁺ basophils (Supplementary Fig. 5c, d). Re-clustering of these three clusters identified 1 cluster of *Fcer1a*⁺*Kit*⁺*Cd34*⁺ pre-BMP-like cells[12], 1 cluster of *Clec12a*hi basophils (Baso1) and 2 clusters of *Clec12a*lo basophils (Baso2 and Baso3) (Fig. 2e, f and Supplementary Fig. 5e, f). Consistent with the flow cytometric analysis data, *Clec12a*hi basophils (Baso1) were abundant in the bone marrow and much fewer in the spleen (Fig. 2e, f). In line with flow cytometric analysis, 21% of *Clec12a*hi Baso1 cells displayed low levels of *Cd34* expression (Fig. 2f and Supplementary Fig. 5f).

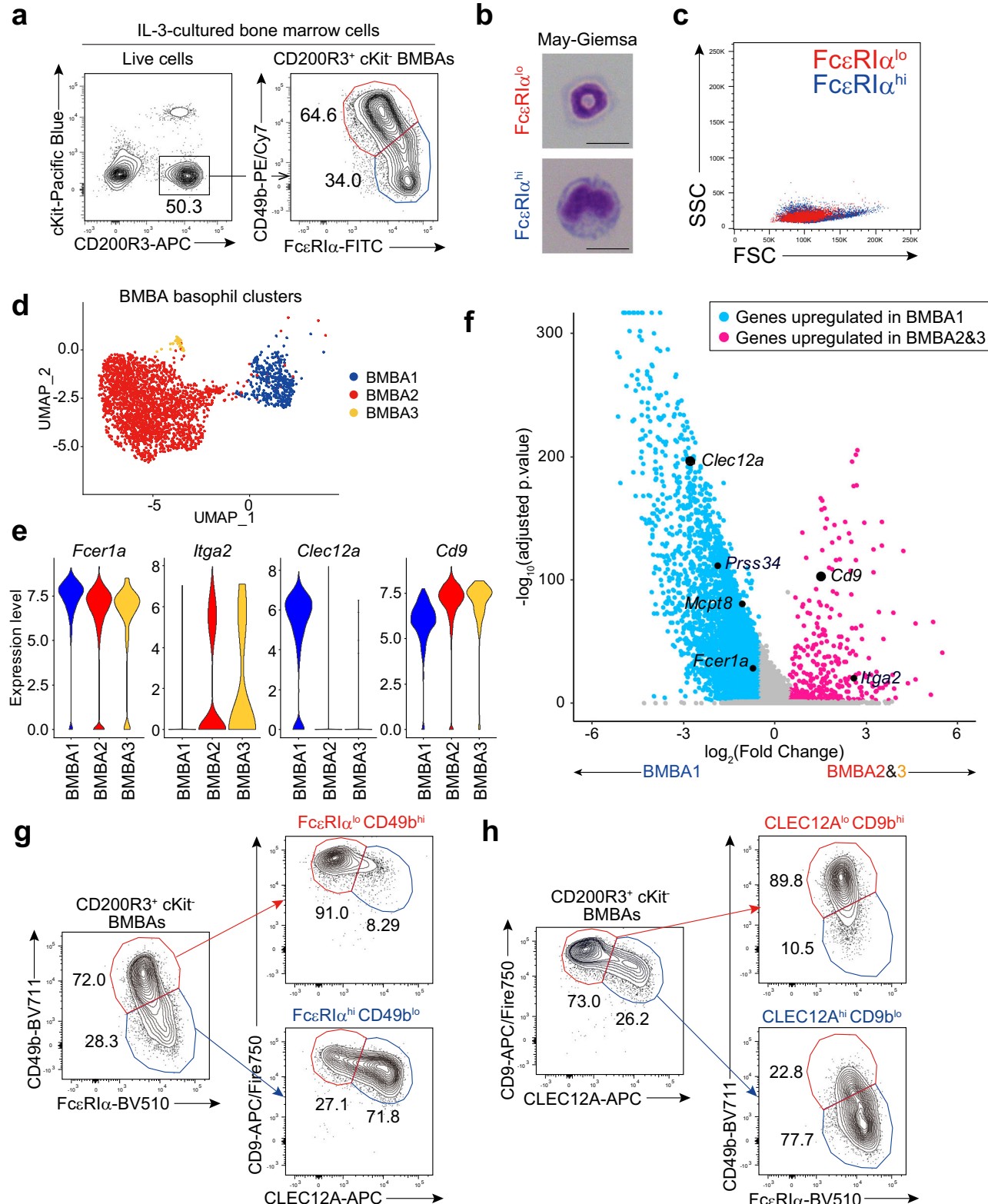

**Fig. 1 | Bone marrow-derived basophils (BMBAs) consist of two distinct sub-populations.** Bone marrow cells were cultured in the presence of 0.3 ng/mL of IL-3 for 7 days. **a** CD200R3⁺cKit⁻ BMBAs were gated (left panel) and the surface expression of FcεRIα and CD49b is shown (right panel). **b**, **c** Two subpopulations of BMBAs (FcεRIαʰⁱCD49bˡᵒ and FcεRIαˡᵒCD49bʰⁱ) were separately isolated and subjected to May-Grünwald-Giemsa staining (Scale bar, 10 μm) (**b**) and flow cytometric analysis (**c**). **d−f** BMBAs were subjected to scRNA-seq analysis. Shown are UMAP plot of basophil clusters (**d**), violin plots of the expression of indicated genes in each cluster (**e**), and volcano plot of genes differentially expressed in Baso1 versus Baso2/3 (**f**). Blue and pink dots indicate genes upregulated in Baso1 and Baso2/3

clusters, respectively. To calculate adjusted *p* value, non-parametric Wilcoxon rank sum test with Bonferonni correction was conducted. **g** CD200R3⁺cKit⁻ BMBAs were subdivided into FcεRIαʰⁱCD49bˡᵒ and FcεRIαˡᵒCD49bʰⁱ subpopulations (left panel), and the surface expression of CLEC12A and CD9 in each subpopulation is shown (right panels). **h** CD200R3⁺cKit⁻ BMBAs were subdivided into CLEC12AˡᵒCD9ʰⁱ and CLEC12AʰⁱCD9ˡᵒ subpopulations (left panel), and the surface expression of FcεRIα and CD49b in each subpopulation is shown (right panels). Data shown in (**a−c**) and (**g**, **h**) are representative of at least three independent experiments. Data in (**d−f**) were obtained from a single experiment.

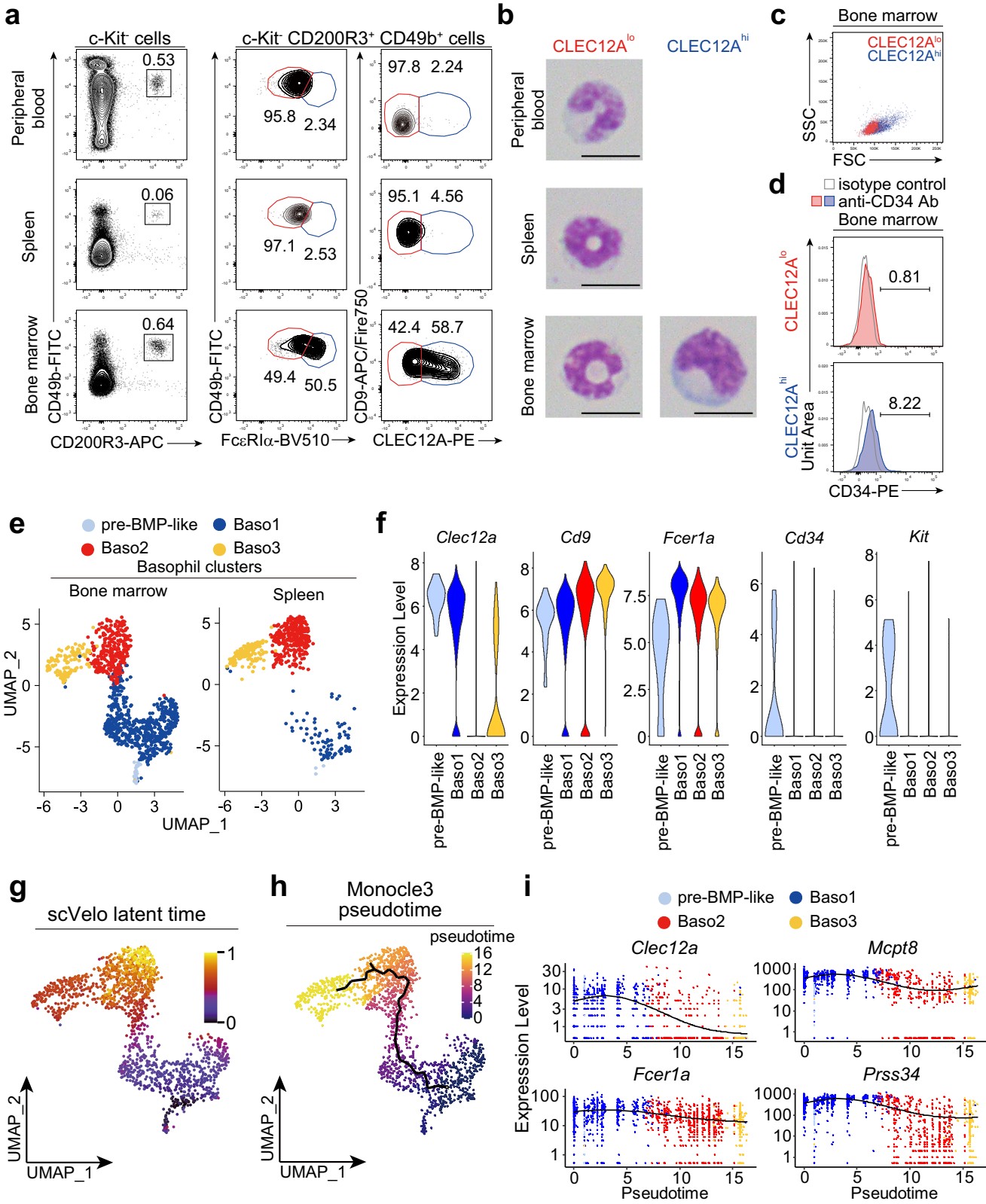

To validate our findings, we re-analyzed the publicly available scRNA-seq datasets reported by Weinreb et al.[26] (GEO accession number: GSE140802) and identified 1 cluster of *Fcer1a*+*Kit*+*Cd34*+ pre-BMP-like cells, 2 clusters of *Clec12a*hi basophils (Baso1 and Baso2) and 3 clusters of *Clec12a*lo basophils (Baso3, Baso4, and Baso5) (Supplementary Fig. 6a–c). A small fraction (11.4%) of *Clec12a*hi basophils displayed low levels of *Cd34* in accordance with the analysis of our scRNA-

seq data, suggesting this small population within *Clec12a*hi basophils may correspond to *Cd34*lo BaP-like cells.

## CLEC12Ahi basophils differentiate into CLEC12Alo basophils ex vivo and in vivo

Pseudo-time trajectory analysis and RNA velocity analysis of our scRNA-seq data inferred the basophil differentiation trajectory from

**Fig. 2 | Two subpopulations of basophils are also detected in the bone marrow while only one of them in the peripheral blood and spleen. a** CD200R3$^+$CD49b$^+$ basophils among the cKit$^-$ population in the peripheral blood, spleen, and bone marrow were gated (left panels), and surface expression of FcεRIα and CD49b (middle panels) and that of CLEC12A and CD9 (right panels) are shown. **b** CLEC12A$^{lo}$CD9$^{hi}$ and CLEC12A$^{hi}$CD9$^{lo}$ subpopulations of basophils in the bone marrow and CLEC12A$^{lo}$CD9$^{hi}$ basophils in the spleen and peripheral blood were separately sort-purified and stained with May-Grünwald Giemsa (scale bar, 10 μm). **c** Scatter plots of CLEC12A$^{hi}$CD9$^{lo}$ (blue dots) and CLEC12A$^{lo}$CD9$^{hi}$ (red dots) subpopulations in the bone marrow are shown. **d** The surface expression of CD34 in each subpopulation is shown. Open histograms indicate control staining with isotype-matched control. **e–i** Basophils were isolated from the bone marrow and spleen of Mcpt8$^{GFP}$ mice (Supplementary Fig. 5) and separately subjected to scRNA-seq analysis. In (**e**), UMAP plots of combined data set of 2745 cells from bone marrow basophils and 731 cells from spleen basophils are shown. In (**f**), violin plots of the expression of indicated genes in each cluster are shown. In (**g**), RNA velocity analysis was conducted on the combined dataset. UMAP plot colored by scVelo latent time is shown. In (**h**), Monocle3 pseudotime analysis was conducted on the combined data set. UMAP plot colored by pseudotime is shown. In (**i**), gene expression changes of indicated genes along with the pseudotime ordering (indicated in Fig. 2h) are shown. Data shown in (**a–d**) are representative of at least three independent experiments. Data in (**e–i**) were obtained from a single experiment.

pre-BMP-like cells to *Clec12a*$^{hi}$ basophils, and then to *Clec12a*$^{lo}$ basophils (Fig. 2g, h and Supplementary Fig. 7). To validate this assumption, we traced changes in surface CLEC12A and CD9 expression when FcεRIα$^{hi}$CD49b$^{lo}$ and FcεRIα$^{lo}$CD49b$^{hi}$ fractions of c-Kit$^-$ basophils were isolated from the bone marrow and separately cultured ex vivo for 2 days. The majority of FcεRIα$^{hi}$CD49b$^{lo}$ basophils changed their surface phenotype from CLEC12A$^{hi}$CD9$^{lo}$ to CLEC12A$^{lo}$CD9$^{hi}$ during the culture (Fig. 3a, left panels). By contrast, FcεRIα$^{lo}$CD49b$^{hi}$ basophils remained CLEC12A$^{lo}$CD9$^{hi}$ even after a 2-day culture (Fig. 3a, right panels). Of note, the total number of cells decreased by ~90% during the 2-day culture of FcεRIα$^{lo}$CD49b$^{hi}$ basophils, whereas the decrease was only ~40% in the culture of FcεRIα$^{hi}$CD49b$^{lo}$ basophils (Fig. 3b, c), suggesting the shorter life span of the former. The compatible results were obtained when we separately cultured the sort-purified CLEC12A$^{hi}$CD9$^{lo}$ and CLEC12A$^{lo}$CD9$^{hi}$ subpopulations of bone marrow basophils and traced changes in surface expression of FcεRIα and CD49b (Supplementary Fig. 8a, b). These results strongly suggest that CLEC12A$^{hi}$CD9$^{lo}$/FcεRIα$^{hi}$CD49b$^{lo}$ basophils are less mature and can differentiate into CLEC12A$^{lo}$CD9$^{hi}$/FcεRIα$^{lo}$CD49b$^{hi}$ mature-type basophils, in spite of the fact that the latter basophils showed lower *Fcer1a*, *Mcpt8* and *Prss34* expression than did the former ones (Fig. 2i and Supplementary Fig. 5f).

We next examined the in vivo relevance of the ex vivo experiments. FcεRIα$^{hi}$CD49b$^{lo}$ and FcεRIα$^{lo}$CD49b$^{hi}$ fractions of c-Kit$^-$ basophils were separately sort-purified from bone marrow cells of CD45.2$^+$ mice and adoptively transferred to sublethally-irradiated CD45.1$^+$ mice (Fig. 3d). As observed in the ex vivo experiments, most of the FcεRIα$^{hi}$CD49b$^{lo}$ basophils shifted their surface phenotype from CLEC12A$^{hi}$CD9$^{lo}$ to CLEC12A$^{lo}$CD9$^{hi}$ in the bone marrow whereas the majority of FcεRIα$^{lo}$CD49b$^{hi}$ basophils remained CLEC12A$^{lo}$CD9$^{hi}$ (Fig. 3d, Supplementary Fig. 9). Taken together, we concluded that CLEC12A$^{hi}$CD9$^{lo}$/FcεRIα$^{hi}$CD49b$^{lo}$ basophils in the bone marrow display precursor-like phenotypes in terms of surface marker expression, cell size and shape of nucleus, and differentiate within the bone marrow into CLEC12A$^{lo}$CD9$^{hi}$/FcεRIα$^{lo}$CD49b$^{hi}$ mature basophils that are also detected in the peripheral blood and spleen. Accordingly, in the following experiments, we designated CLEC12A$^{hi}$CD9$^{lo}$/FcεRIα$^{hi}$CD49b$^{lo}$ and CLEC12A$^{lo}$CD9$^{hi}$/FcεRIα$^{lo}$CD49b$^{hi}$ basophils as pre-basophils and mature basophils, respectively.

### Pre-basophils and mature basophils display distinct properties of cell proliferation and activation

To further characterize pre-basophils and mature basophils, we isolated CLEC12A$^{hi}$CD9$^{lo}$CD34$^-$ pre-basophils from the bone marrow as well as CLEC12A$^{lo}$CD9$^{hi}$ mature basophils from the bone marrow and spleen separately and compared their gene expression profiles using bulk RNA-seq analysis (Fig. 4a). As a reference, we also included sort-purified CD34$^+$ BaPs that has been defined as uni-potent basophil progenitors. Hierarchical clustering analysis of DEGs identified that the gene expression profiles of mature basophils isolated from the bone marrow and spleen resembled each other whereas CD34$^-$ pre-basophils and mature basophils showed distinct gene expression profiles (Fig. 4a). Of note, CD34$^-$ pre-basophils displayed a gene

expression profile similar to that of CD34$^+$ BaPs (Fig. 4a), strengthening our forementioned prediction that CD34$^+$ BaPs are included in the CLEC12A$^{hi}$ pre-basophil population in terms of the gene expression profile.

Gene ontology (GO) enrichment analysis showed that GO terms associated with cell proliferation, such as "nuclear division", "chromosome segregation", and "DNA replication" were enriched in CLEC12A$^{hi}$ pre-basophils when compared to CLEC12A$^{lo}$ mature basophils (Fig. 4b), in accordance with the scRNA-seq analysis showing that *Clec12a*$^{hi}$ Baso1 cluster displayed high S phase score, compared with *Clec12a*$^{lo}$ Baso2/3 clusters (Supplementary Fig. 10). Indeed, the uptake of 5-Ethynyl-2′-deoxyuridine (EdU) nucleotide analogue was clearly detected in pre-basophils but not mature basophils (Fig. 4d), indicating the proliferation capacity of pre-basophils. This difference in the capability of proliferation between pre-basophils and mature basophils appears to explain the difference in the reduction of the total cell numbers observed during the 2-day culture (Fig. 3c).

Mature basophils were enriched with GO terms associated with immune effector functions such as "regulation of immune effector process" and "positive regulation of cytokine production" when compared to pre-basophils (Fig. 4c). When stimulated with IgE plus antigens, CLEC12A$^{lo}$ mature basophils showed higher levels of CD63 expression (an indicator of degranulation) and IL-4 production than did CLEC12A$^{hi}$ pre-basophils (Fig. 4e, f), although mature basophils expressed lower levels of surface FcεRI expression (Fig. 2a). Intriguingly, when stimulated with innate-type stimuli such as IL-3, IL-33 and LPS, pre-basophils produced higher levels of IL-4 than did mature ones (Fig. 4g). Furthermore, the bulk RNA-seq analysis identified that IL-3-stimulated pre-basophils display gene expression profiles distinct from those of IL-3- or antigen/IgE-stimulated mature basophils, including upregulated expression of *Il10* and *Il13* in the former (Supplementary Fig. 11a, b). Thus, pre-basophils appeared to be more reactive to non-IgE stimulation whereas mature basophils are more reactive to IgE/allergen stimulation.

Two populations of basophils with distinct phenotypic and functional properties were reported, depending on the cytokine milieu, namely the presence of either IL-3 or TSLP, during the generation of basophils[27]. When stimulated with IL-3 and IL-33, TSLP-elicited basophils produce more IL-4 and IL-6 compared to IL-3-elicited basophils. In contrast, TSLP-elicited basophils are less responsive to IgE cross-linking in terms of degranulation. This reactivity of TSLP-elicited basophils is apparently similar to that of pre-basophils identified in the present study. However, TSLP-elicited basophils are relatively small in size and possess ring-like nuclei characteristic to mature-type basophils[27]. In accordance with this, we found that most TSLP-elicited basophils display the FcεRIα$^{lo}$CD49b$^{hi}$/CLEC12A$^{lo}$CD9$^{hi}$ phenotype corresponding to mature-type basophils (Supplementary Fig. 12a). Moreover, TSLP-elicited mature basophils displayed a gene expression profile similar to that of IL-3-elicited mature rather than pre-basophils (Supplementary Fig. 12b). Taken together, TSLP-elicited basophils do not seem to correspond to pre-basophils and can be categorized as a subpopulation of mature basophils.

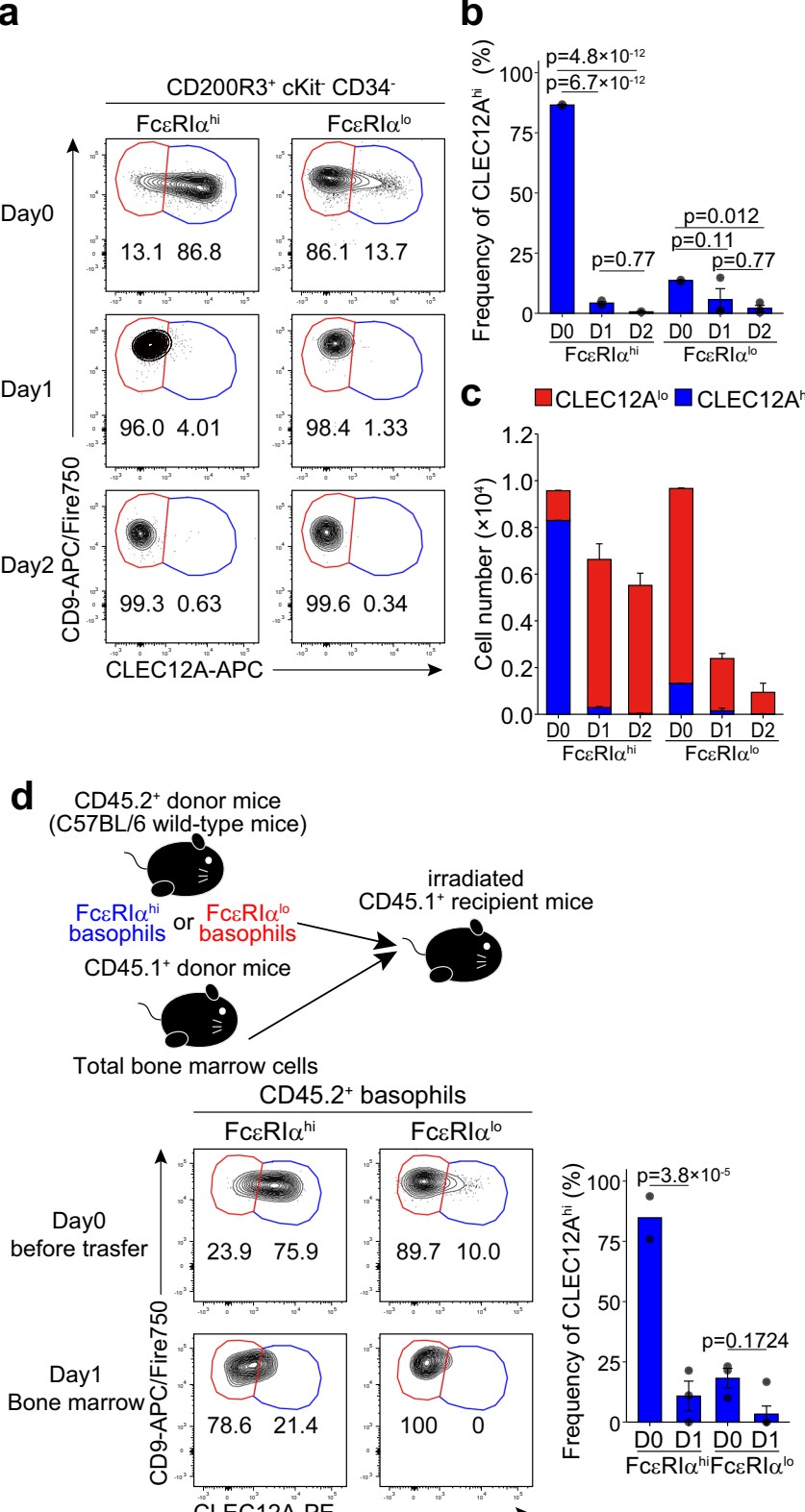

## Not only mature but also pre-basophils emerge in the peripheral tissues during helminth infection

Basophils are often shown to infiltrate peripheral tissues during parasitic infections[8]. *Nippostrongylus brasiliensis* (Nb) is a widely studied helminth in mouse models of human hookworm infection. They enter host animals through the skin and migrate to the lungs and finally to the intestine. After worms have passed through the lungs, basophils

accumulate in the lungs and are crucial for the suppression of exaggerated lung inflammation and repair of damaged lung tissue[28]. On day 7 post Nb-infection, we detected CLEC12A^hi pre-basophils, in addition to CLEC12A^lo mature basophils, in the spleen, blood and even lungs (Fig. 5a), in contrast to the observation that pre-basophils largely reside within the bone marrow in uninfected or PBS-injected control mice (Fig. 2a and Supplementary Fig. 13a, b). Pre-basophils isolated

**Fig. 3 | CLEC12A^{hi}CD9^{lo} basophils differentiate into CLEC12A^{lo}CD9^{hi} basophils ex vivo and in vivo.** **a**–**c** FcεRIα^{hi}CD49b^{lo} and FcεRIα^{lo}CD49b^{hi} subpopulations of basophils sort-purified from the bone marrow were separately cultured ex vivo for 1 or 2 days. In (**a**), the time course of their surface CLEC12A and CD9 expression is shown. In (**b**), the change in the frequency of CLEC12A^{hi} cells during the culture is shown (mean ± SEM, n = 3 each). In (**c**), the change in the number of live cells during the culture is shown. The red and blue bars correspond to the CLEC12A^{lo}CD9^{hi} and CLEC12A^{hi}CD9^{lo} fractions, respectively (mean ± SEM, n = 3 each). Data in (**a**–**c**) are representative of at least three independent experiments. **d** FcεRIα^{hi}CD49b^{lo} and FcεRIα^{lo}CD49b^{hi} subpopulations of basophils were separately isolated from the bone marrow of CD45.2^+ C57BL/6 mice (5 × 10^4 cells/mouse), mixed with CD45.1^+ whole bone marrow cells (0.3 × 10^6 cells/mouse), and intravenously administered to sublethally-irradiated CD45.1^+ congenic C57BL/6 mice. One day after the cell transfer, bone marrow cells were prepared from recipient mice and subjected to flow cytometric analysis for the expression of CLEC12A and CD9 on CD45.2^+ donor-derived basophils (bottom panels). For comparison, the CLEC12A and CD9 expression on transferred cells (Day0) is displayed in the upper panels. The frequency of CLEC12A^{hi} cells before and 1 day after the transfer in each experimental group is shown in the right panel where data are pooled from two experiments (mean ± SEM, n = 2 for FcεRIα^{hi} on D0, n = 3 for FcεRIα^{lo} on D0, n = 3 for FcεRIα^{hi} on D1, n = 5 for FcεRIα^{lo} on D1). Two-way ANOVA with Tukey's multiple comparisons test was used for multiple comparisons (**b**, **d**).

from Nb-infected lungs displayed higher ex vivo EdU incorporation than did mature basophils (Fig. 5b), suggesting that pre-basophils retain their high proliferative capacity even outside of the bone marrow. Moreover, pre-basophils accumulating in the infected lungs showed IL-4 expression comparable to that of mature basophils, implying their possible contribution to Th2 immunity against Nb infection (Fig. 5c).

We previously demonstrated that in the second Nb infection, basophils infiltrated the infected skin and trapped worms within the skin to prevent further spread of infection to the lungs and intestines[29]. CLEC12A^{hi} pre-basophils besides CLEC12A^{lo} mature ones were detected in skin lesions during the second Nb infection (Fig. 5d). The scRNA-seq analysis of CD200R3^+cKit^- cells isolated from the bone marrow and the Nb-infected skin of twice-infected mice identified 9 clusters, among which clusters 0, 1, and 2 corresponded to *Cd200r3^+Mcpt8^+* basophils (Supplementary Fig. 14a, b). Among these three basophil clusters, we detected one cluster of *Clec12a*^{hi} pre-basophils and two clusters of *Clec12a*^{lo} mature basophils in both organs (Fig. 5e, f and Supplementary Fig. 14c). Integrated analysis of scRNA-seq data of basophils from Nb-infected and uninfected mice identified that pre-basophils detected in the skin of Nb-infected mice display gene expression profiles close to those of pre-basophils in the bone marrow of uninfected mice (Supplementary Fig. 15), including upregulated expression of *Fcer1a*, *Clec12a*, *Mcpt8* and *Prss34* (Fig. 5f). Moreover, gene set enrichment analysis (GSEA) and Gene Ontology enrichment analysis uncovered that pre-basophils in the Nb-infected skin display the enrichment of genes associated with cell proliferation, including "chromosome segregation" (Fig. 5g, Supplementary Fig. 16), suggesting that pre-basophils retain their high proliferative capacity in the infected skin as observed in the infected lungs.

### IL-3-mediated CXCR4 downregulation in pre-basophils promotes their egress from the bone marrow

We next sought to clarify a possible mechanism underlying the emergence of pre-basophils in peripheral tissues during Nb infections. Nb infections have been shown to induce systemic upregulation of IL-3[30], and therefore we examined the possible contribution of IL-3 to the emergence of pre-basophils outside of the bone marrow. Intraperitoneal administration of IL-3 complexes (recombinant IL-3 mixed with monoclonal anti-IL-3 antibody) in mice resulted in the appearance of CLEC12A^{hi} pre-basophils in the peripheral blood (Fig. 6a). We postulated that IL-3 induced the egress of pre-basophils from the bone marrow. To address this possibility, we conducted bulk RNA-seq analysis of IL-3- and control PBS-treated BMBAs and identified the DEGs between them. We particularly took notice of *Cxcr4* whose expression was significantly decreased by the IL-3 stimulation (Fig. 6b), because CXCR4 has been shown to be an essential chemokine receptor for the retention of immune cells in the bone marrow[31,32]. The scRNA-seq and flow cytometric analyses of bone marrow cells demonstrated that the expression of CXCR4 was higher in pre-basophils than in mature basophils at both transcript and protein levels (Fig. 6c, Supplementary Fig. 17), and that IL-3 stimulation reduced surface CXCR4 expression in both CLEC12A^{lo} mature and CLEC12A^{hi} pre-basophils (Fig. 6d). These

results suggested that pre-basophils remain within the bone marrow owing to their high expression of CXCR4 and that CXCR4 down-regulation by IL-3 may promote the egress of pre-basophils from the bone marrow. In accordance with this assumption, the treatment of mice with intraperitoneal administration of AMD3100, a CXCR4 inhibitor, resulted in the emergence of CLEC12A^{hi} pre-basophils in the peripheral blood and spleen (Fig. 6e). Thus, IL-3-mediated down-regulation of CXCR4 in pre-basophils in the bone marrow appeared to account, at least in part, for the egress of pre-basophils from the bone marrow and their appearance in the lungs and skin during Nb infections.

## Discussion

Advancements of scRNA-seq technology have enabled us to understand the differentiation trajectory of various hematopoietic cell lineages and revisited the classical view of hematopoietic cell differentiation. In the present study, we took advantage of this technology combined with flow cytometric and functional analyses and succeeded in the identification of CLEC12A^{hi}CD9^{lo} pre-basophils that are developmentally located downstream of pre-basophil and mast cell progenitors (pre-BMPs) and upstream of CLEC12A^{lo}CD9^{hi} mature basophils in mice. The transcriptomic analysis predicted that the pre-basophil population include previously-defined basophil-progenitor (BaP)-like cells in terms of the gene expression profile.

We found that pre-basophils in the bone marrow are highly proliferative in contrast to mature basophils. This is also the case in pre-basophils accumulating in Nb-infected peripheral tissues. Even though the frequency of pre-basophils is much less than that of mature ones in the infected tissues, we assume that their proliferation and differentiation to mature basophils as well as their continuous accumulation in the infected tissues enable the efficient expansion of basophils for the protection against parasitic infection, as in the case of bacterial infections that induce the emergence of immature neutrophils in peripheral tissues[33]. Besides the proliferative capacity, pre-basophils were found to display immune responsiveness distinct from that of mature basophils. As expected from their location at the earlier differentiation stage, pre-basophils were less activated in response to IgE-mediated stimulation than did mature ones, even though they express higher levels of FcεRI on the cell surface. Intriguingly, however, pre-basophils responded more vigorously than mature ones to IgE-independent stimulation such as IL-3 and IL-33 in terms of IL-4 secretion. Of note, pre-basophils accumulating in the Nb-infected lungs showed IL-4 expression comparable to that of mature basophils, suggesting their possible contribution to Th2 immunity against Nb infection. The transcriptomic analysis in the present study identified that IL-3-stimulated pre-basophils display higher expression of *Il13* and *Il10* compared to mature basophils. Considering that IL-10 and IL-13 dampen tissue damage in the lungs during Nb infection[34,35], pre-basophils may exert regulatory functions in Nb-infected tissues. Interestingly, Rodriguez Gomez et al. reported the emergence of FcεRI^{++} basophils in the spleen of colitis model mice where FcεRI^{++} basophils produced significantly higher amounts of IL-4 and IL-6 in response to IL-3 than did FcεRI^+ basophils[36]. These characteristics of

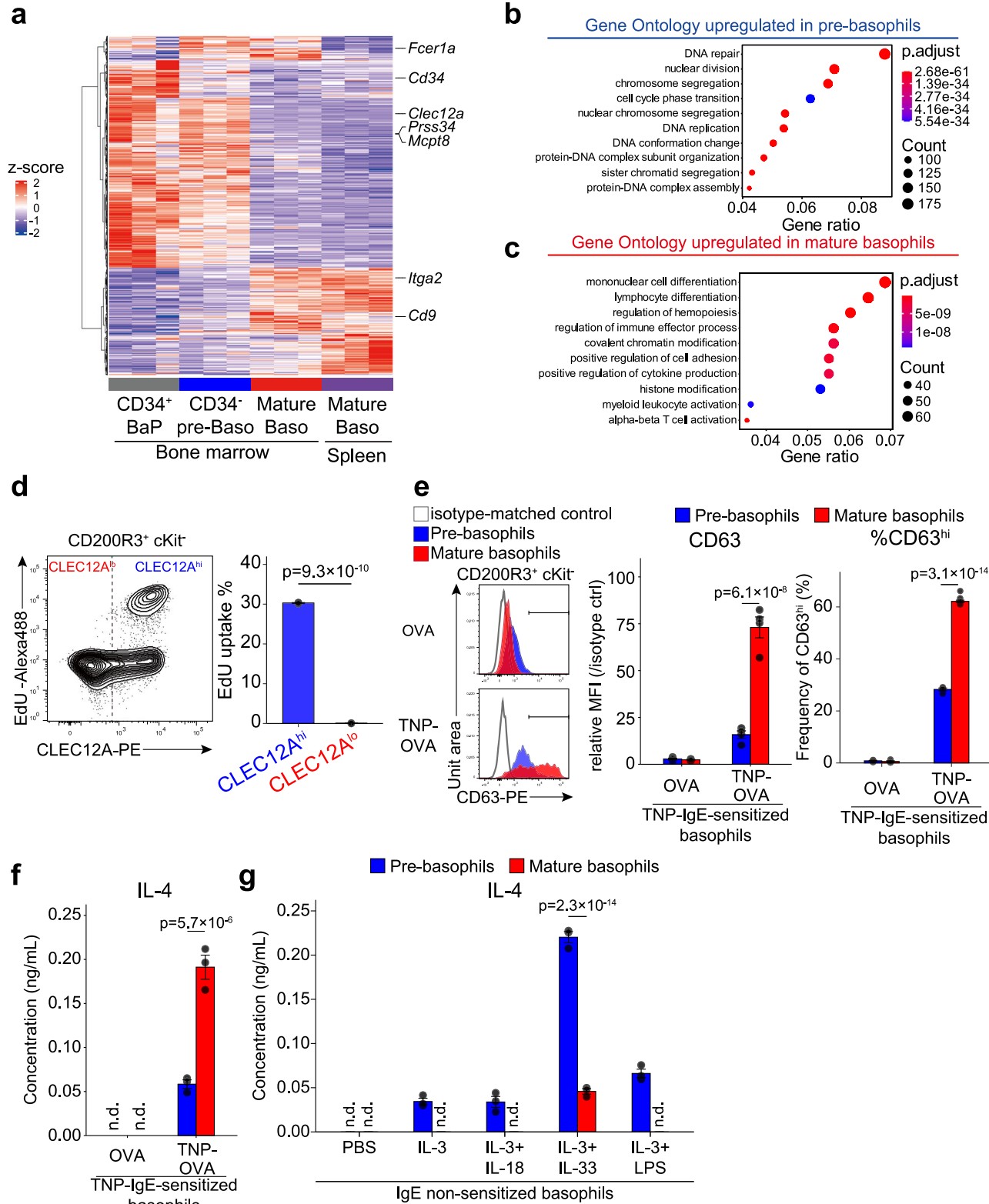

FcεRI++ basophils are consistent with those of pre-basophils identified in the present study. Basophil-derived IL-4 and IL-6 were shown to dampen inflammation in the colon[36], suggesting the possible contribution of FcεRI++ pre-basophils to the regulation of colitis.

The interaction of CXCR4 and its ligand CXCL12 regulates the physiological distribution of HSCs and neutrophils[37]. Under homeostatic conditions, CXCL12-expressing stromal cells retain CXCR4-expressing neutrophils in the bone marrow whereas inflammatory

cues such as G-CSF promote the egress of neutrophils, including immature ones, by down-regulating the surface expression of CXCR4, leading to the emergence of immature neutrophils in the circulation[37]. Although CXCR4 on basophils has been reported to be involved in the migration toward the inflamed skin and secondary lymphoid organs[38,39], its function in the basophil egress from the bone marrow remained unexplored. In the present study, we demonstrated that the inhibition of CXCR4 by AMD3100 promotes the egress of pre-

**Fig. 4 | CLEC12A^hi CD9^lo pre-basophils and CLEC12A^lo CD9^hi mature basophils display distinct properties of cell proliferation and activation. a–c** CD34^+ BaPs, CD34^- CLEC12A^hi CD9^lo pre-basophils (CD34^- pre-Baso) and CLEC12A^lo CD9^hi mature basophils (Mature Baso) isolated from the bone marrow as well as CLEC12A^lo CD9^hi mature basophils isolated from the spleen were subjected to bulk RNA-seq analysis (*n* = 3, pooled from 4 animals for each replicate). In (**a**), the hierarchically clustered heatmap of DEGs is shown. In (**b** and **c**), the top 10 enriched GO terms upregulated in pre-basophils and in mature basophils, respectively, are plotted in order of gene ratio. The size of the dots indicates the number of genes associated with indicated GO terms while the color of the dots indicates the adjusted *P* values (p. adjust) calculated by one-sided Fisher's exact test with Benjamini-Hochberg correction. **d** CD49b^+ cells isolated from the bone marrow were incubated ex vivo with EdU for 2 h. EdU uptake in CLEC12A^hi and CLEC12A^lo basophils is shown. **e** and **f** Mice were intravenously sensitized with anti-TNP IgE 1 day prior to the experiment. CLEC12A^hi and CLEC12A^lo basophils isolated from the bone marrow of TNP-IgE-sensitized mice were separately cultured ex vivo in the presence of TNP-conjugated OVA (TNP-OVA) or control OVA for 1 h (**e**) or 6 h (**f**). In (**e**), the surface CD63 expression on CLEC12A^hi (blue shaded histogram) and CLEC12A^lo (red shaded histogram) basophils is shown in the left panels (mean ± SEM, *n* = 4 each). The ratio of MFI, namely MFI of CD63 staining divided by MFI of control staining, is shown in each experimental group (middle panel, mean ± SEM, *n* = 4 each). The frequency of CD63^hi cells in each experimental group is shown (right panel, mean ± SEM, *n* = 4 each). In (**f**), IL-4 concentration in culture supernatants in each experimental group is shown (mean ± SEM, *n* = 3 each). **g** CLEC12A^hi and CLEC12A^lo basophils isolated from the bone marrow were separately cultured ex vivo in the presence of PBS, IL-3 alone, IL-3 + IL-18, IL-3 + IL-33 or IL-3 + LPS for 6 h. The IL-4 concentration in culture supernatants is shown (mean ± SEM, *n* = 3 each). Data in (**a**–**c**) were obtained from a single experiment. Those shown in (**d**–**g**) are representative of at least three independent experiments. n.d.; not detected. Unpaired Student's *t* test was used for comparing two groups (**d**) and two-way ANOVA with Tukey's multiple comparisons test for multiple comparisons (**e**–**g**).

basophils from the bone marrow. Furthermore, the ex vivo stimulation with IL-3 reduced the expression of CXCR4 on pre-basophils, suggesting the possibility that the systemic upregulation of IL-3 leads to the egress of pre-basophils from the bone marrow. A similar mechanism may operate during Nb infection to promote the emergence of pre-basophils in peripheral tissues.

In conclusion, we identified c-Kit^- CLEC12A^hi pre-basophils that are located downstream of pre-BMPs and upstream of mature basophils in the basophil ontogeny. Pre-basophils display the proliferative capacity and activation property distinct from those of mature basophils. While pre-basophils are retained in the bone marrow under homeostatic conditions, they appear in peripheral tissues during helminth infection and display proliferative and activating properties. Taken together, the present study identified pre-basophils that bridge the gap between pre-BMPs and mature basophils along the basophil differentiation pathway in mice. Human basophils have been shown to express CLEC12A[40] and therefore further studies are warranted to identify the human counterpart of mouse pre-basophils.

## Methods

### Mice
C57BL/6 J (C57BL/6 JmsSlc) and BALB/c (BALB/cCrSlc) mice were purchased from Japan SLC. *Mcpt8*^GFP transgenic C57BL/6 mice (B6-Tg(*Mcpt8*-GFP)^HKar) were developed in our laboratory[4]. IL-4 reporter G4 (*Il4*^GFP/+) C57BL/6 mice (B6.129P2-*Il4*^tm1.1Wep) were provided by Dr William E Paul[41]. CD45.1 congenic C57BL/6 mice (B6.SJL-*Ptprc*^a *Pepc*^b/BoyJ) were purchased from the Jackson Laboratory. In most of the experiments except for Supplementary Fig. 2, C57BL/6J background mice were used. In Supplementary Fig. 2a–d, BALB/c mice were used. Mice were maintained under specific pathogen-free conditions in our animal facilities. Animal rooms are maintained at 22 ± 3 °C, with a 12 h:12 h light-dark cycle, and humidity is maintained between 30% and 70%. Animals are housed in individually ventilated cages (Innovive Caging System) and are fed with natural ingredient chow diet ad libitum (Japan CLEA: Rodent Diet CE-2). Cages are bedded with cloth material (Japan SLC: Q-pura chip). Animals are group housed whenever possible. At the end of all experiments, mice were euthanized by CO_2 Inhalation. All animal studies were approved by the Institutional Animal Care and Use Committee of Tokyo Medical and Dental University (No. A2022-023C).

### Antibodies
The following antibodies were purchased from BioLegend: biotinylated anti-B220 (clone: RA3-6B2, catalog #: 103204, dilution 1:400), anti-CD3ε (clone: 145-2C11, catalog #: 100304, dilution 1:400), anti-CD4 (clone: GK1.5, catalog #: 100404, dilution 1:400), anti-CD8a (clone: 53-6.7, catalog #: 100704, dilution 1:400), anti-CD19 (clone: 6D5, catalog #: 115504, dilution 1:400), anti-CD49b (clone: DX5, catalog #: 108904, dilution 1:400), anti-Gr-1 (clone: RB6-8C5, catalog #: 108404, dilution 1:400), and anti-TER-119 (clone: TER-119, catalog #: 116204, dilution 1:400); FITC-conjugated anti-CD49b (clone: HMα2, catalog #: 103504, dilution 1:400), anti-FcεRIα (clone: MAR-1, catalog #: 134306, dilution 1:400), and anti-CD45 (clone: 30-F11, catalog #: 103108, dilution 1:400); Alexa Fluor 488-conjugated anti-CD45.2 (clone: 104, catalog #: 109816, dilution 1:400); PE-conjugated anti-CD34 (clone: SA376A4, catalog #: 152204, dilution 1:400), anti-CD63 (clone: NVG-2, catalog #: 143904, dilution 1:400), anti-CLEC12A (clone: 5D3/CLEC12A, catalog #: 143404, dilution 1:400), anti-CXCR4 (clone: L276F12, catalog #: 146506, dilution 1:400), rat IgG2a, κ Isotype Ctrl (clone: RTK2758, catalog #: 400508, dilution 1:400) and rat IgG2b,κ Isotype Ctrl (clone: RTK4530, catalog #: 400608, dilution 1:400); PE-Cy7-conjugated anti-CD49b (clone: HMα2, catalog #: 103518, dilution 1:400) and anti-CD200R3 (clone: Ba13, catalog #: 142209, dilution 1:400); APC-conjugated anti-CD200R3 (clone: Ba13, catalog #: 142208, dilution 1:400), anti-Sca-1 (clone: D7, catalog #: 108112, dilution 1:400) and anti-CLEC12A (clone: 5D3/CLEC12A, catalog #: 143406, dilution 1:400); APC/Cy7-conjugated anti-CD16/32 (clone: S17011E, catalog #: 156612, dilution 1:400); APC/Fire750-conjugated anti-CD9 (clone: MZ3, catalog #: 124814, dilution 1:400); Pacific Blue-conjugated anti-c-Kit (clone: 2B8, catalog #: 105820, dilution 1:400) and anti-CD45.1 (clone: A20, catalog #: 110722, dilution 1:400); BV421-conjugated anti-CD34 (clone: SA376A4, catalog #: 152208, dilution 1:400) and anti-CD45 (clone: 30-F11, catalog #: 103134, dilution 1:400); BV510-conjugated anti-FcεRIα (clone: MAR-1, catalog #: 134327, dilution 1:400); BV605-conjugated anti-c-Kit (clone: 2B8, catalog #: 105847, dilution 1:400); BV711-conjugated anti-CXCR4 (clone: L276F12, catalog #: 146517, dilution 1:400); BV786-conjugated anti-IL-7Rα (clone: A7R34, catalog #: 135037, dilution 1:400); PerCP/Cy5.5-conjugated anti-TER119 (clone: TER-119, catalog #: 116228, dilution 1:400). BD OptiBuild BV711-conjugated anti-CD49b (clone: HMα2, catalog #: 740704, dilution 1:400) was purchased from BD Biosciences. FITC-conjugated Streptavidin (catalog #: 11-4317-87, dilution 1:400) was purchased from Thermo Fisher Scientific. TNP-specific IgE antibody (IgE-Lb4, purchased from ADCC) was prepared in our laboratory.

### Cell culture and the generation of BMBAs
Cell culture was performed in complete RPMI (RPMI 1640 medium (Nacalai tesque) supplemented with 10% FBS (Sigma), 2mM L-glutamine (Nacalai tesque), 1 mM sodium pyruvate (Nacalai tesque), 0.055 mM 2-mercaptoethanol (Gibco), 100 U/mL penicillin/streptomycin (Nacalai tesque), and 0.1 mM MEM non-essential amino acids solution (Nacalai tesque)). Bone marrow-derived basophils (BMBAs) were generated by culturing bone marrow cells in the presence of varying concentrations of mouse IL-3 (0–100 ng/mL: BioLegend) for 7 d. For most of the experiments except for Supplementary Fig. 2, 0.3 ng/mL of IL-3 were used for generating BMBAs.

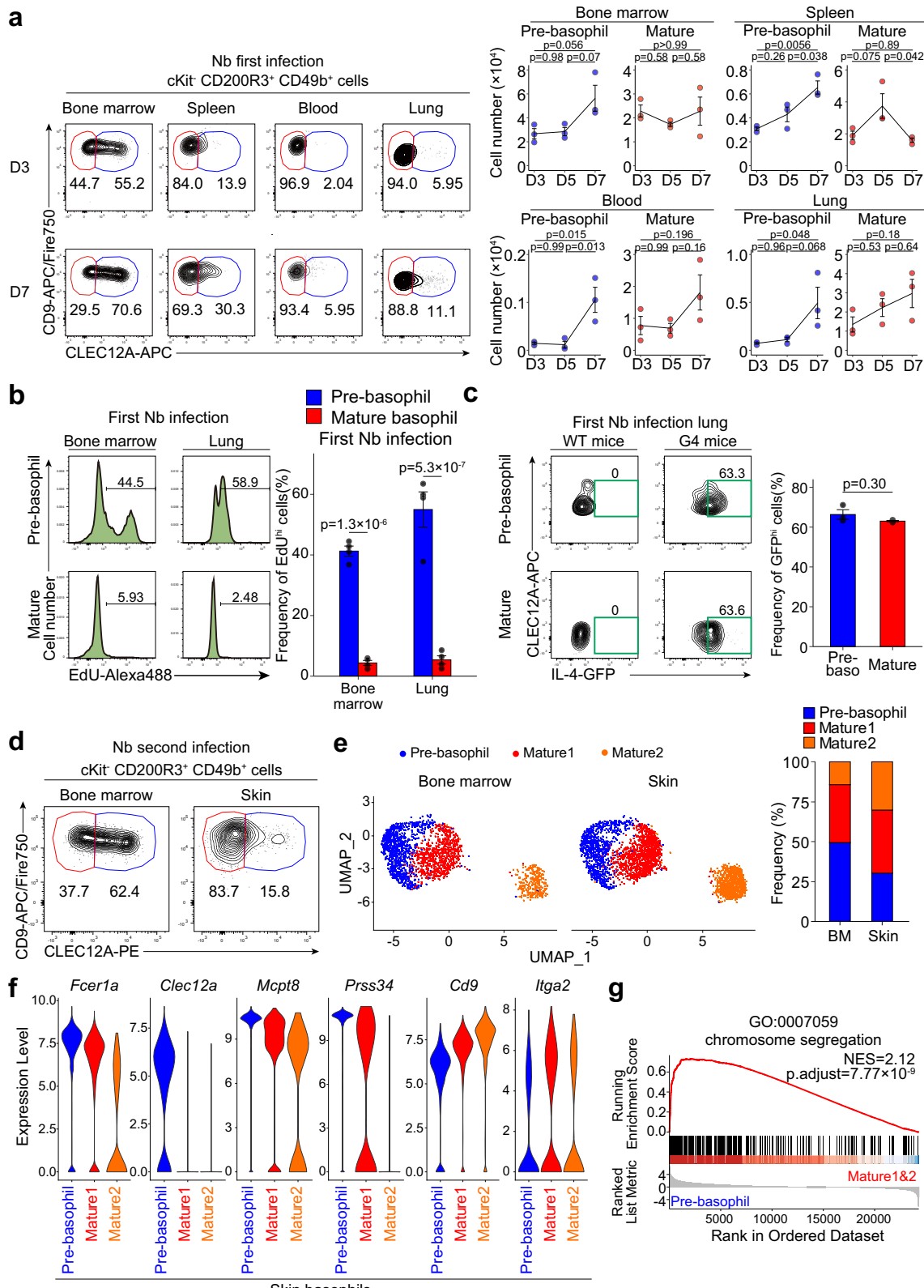

**Flow cytometric analysis and cell sorting**

Single cell suspensions were prepared from the bone marrow and spleen, followed by red blood cell (RBC) lysis by using RBC Lysis Buffer (BioLegend) for 5 min. For the flow cytometric analysis of peripheral blood, leukocytes from the heparinized blood sample were isolated from the interphase of 25% and 65% Percoll PLUS (cytiva) after gradient centrifugation. For the flow cytometric analysis of lung and skin

samples, tissues were minced by razors and dissociated by using gentleMACS dissociator (Miltenyi Biotec) and Lung Dissociation Kit (Miltenyi Biotec, catalog #: 130-095-927) or Multi Tissue Dissociation Kit 1 (Miltenyi Biotec, catalog #: 130-110-201). Cells were stained with the indicated antibodies after treatment with normal rat serum (Merck Millipore) and TrueStain FcX PLUS (2.5 μg/mL; BioLegend; clone: S17011E, catalog#: 156604, dilution 1:200) to prevent non-specific

**Fig. 5 | Pre-basophils can be detected in peripheral tissues during helminth infection. a** C57BL/6 mice were infected with Nb larvae, and cells were prepared from the bone marrow, spleen, peripheral blood and lungs on days 3, 5 and 7 post-infection. The expression of CLEC12A and CD9 on cKit⁻CD200R3⁺CD49b⁺ basophils in each time point (left panels) and the time course of cell numbers of CLEC12A^hi pre-basophils and CLEC12A^lo basophils (Mature) in each experimental group (right panels) are shown (mean ± SEM, *n* = 3 each). **b** CD49b⁺ cells isolated from the bone marrow and lungs on day 7 post Nb-infection were incubated ex vivo with EdU for 2 h. EdU uptake in pre-basophils and mature basophils is shown in the left panels. The frequency of EdU⁺ cells is shown in the right panel (mean ± SEM, *n* = 3 each). **c** WT mice and IL-4 reporter (G4) mice were infected with Nb larvae. GFP fluorescence in pre-basophils (Pre-baso) and mature basophils on day 7 post Nb-infection is shown in the left panels. MFI of GFP is shown in the right panel (mean ± SEM, *n* = 3 each). **d–g** C57BL/6 mice were infected twice with Nb larvae, and on day 2 of the second infection cKit⁻CD200R3⁺CD49b⁺ basophils were isolated from the bone

marrow and infected skin. In (**d**), their surface expression of CLEC12A and CD9 is shown. **e–g** They were separately subjected to scRNA-seq analysis. In (**e**), UMAP plots show 3 basophil clusters (pre-basophil, Mature 1 and 2; defined by Seurat clustering) in the left panel. The frequency of each cluster in the bone marrow (BM) and skin is shown in the right panel. In (**f**), violin plots of the expression of indicated genes in each cluster are shown. In (**g**), the gene set enrichment analysis comparing pre-basophil and mature 1&2 was conducted. GSEA enrichment plot of genes involved in the chromosome segregation is shown. Normalized enrichment scores (NES) and Benjamini-Hochberg-adjusted *p* values (p.adjust) calculated by a permutation test are shown. Data shown in (**a–d**) are representative of at least three independent experiments. Data in (**e–g**) were obtained from a single experiment. Unpaired Student's *t* test was used for comparing two groups (**c**) and one-way ANOVA with Tukey's multiple comparisons test (**a**) or two-way ANOVA with Tukey's multiple comparisons test (**b**) for multiple comparisons.

binding. Stained cells were analyzed with FACSLyric (BD Biosciences) and FlowJo software ver 10.8.1 (BD Biosciences) or sorted with FACS AriaIII (BD Biosciences). For ex vivo EdU uptake assay, cells collected from the bone marrow or lung were cultured for 2 h with 10 µM EdU (5-ethynyl-2′-deoxyuridine). After culture, EdU staining was performed by using Click-iT Plus EdU flow cytometry assay kits (ThermoFisher Scientific, catalog #: C10633) according to the manufacturer's protocol.

## Giemsa staining
Cytospin samples of sort-purified basophils were stained with May-Grünwald and Giemsa stain (FUJIFILM Wako Pure Chemical).

## Sample preparation for scRNA-seq analysis
For scRNA-seq in Fig. 1, BMBAs generated by culturing bone marrow cells in the presence of IL-3 (0.3 ng/mL: BioLegend) for 7 days were subjected to scRNA-seq analysis. BMBAs generated from 5 mice were pooled to generate scRNA-seq data. For scRNA-seq in Fig. 2, single-cell samples were prepared from the bone marrow and spleen of *Mcpt8*^GFP transgenic mice and depleted of lineage-positive cells by using biotinylated lineage antibodies (anti-CD4, anti-CD8a, anti-CD19, anti-B220, anti-Gr-1 and anti-TER-119, dilution 1:400) and EasySep Mouse Streptavidin RapidSpheres Isolation Kit (Stemcell Technologies, catalog #: ST-19860). GFP-positive cells were sort-purified from lineage-negative cells. Basophils isolated from 5 mice were pooled to generate scRNA-seq data. For scRNA-seq in Fig. 5, single-cell suspensions were prepared from the bone marrow and infected skin and subjected to positive selection of CD49b-positive cells by using biotinylated anti-CD49b antibody (BioLegend; clone: DX5, catalog #: 108904, dilution 1:400) and EasySep mouse Biotin Positive Selection Kit II (Stemcell Technologies, catalog #: ST-17665) followed by the isolation of CD200R3⁺cKit⁻ cells by cell sorting. Basophils isolated from 5 mice were pooled to generate scRNA-seq data.

## Data processing for scRNA-seq analysis
Single cell suspensions were reacted with 2.5 µg/mL of Totalseq anti-mouse Hashtag-A antibodies [BioLegend, clone: M1/42, 30-F11, dilution 1:400; Hashtag antibody A0301 (catalog #: 155801), A0302 (catalog #: 155803), and A0306 (catalog #: 155811) are used for scRNA-seq in Fig. 1 and Fig. 2; A0312 (catalog #: 155823) and A0313 (catalog #: 155825) are used for scRNA-seq in Fig. 5]. Obtained single-cell suspensions were subjected to a BD Rhapsody system with BD Rhapsody Targeted & Abseq Reagent kit (BD Biosciences, catalog#: 633771) following the manufacturer's instructions. Resultant beads were reverse transcribed, and treated with Exonuclease I for 60 min. The reverse-transcribed, Exonuclease I-treated BD Rhapsody beads were subjected to TAS-Seq workflow[42] for cDNA and hashtag library amplification. Briefly, BD Rhapsody beads were subjected to TdT (Roche, catalog #: 3333574001) and RNase H (Enzymatics, catalog #: Y9220F) reactions, followed by second-strand synthesis and 1st round of

whole-transcriptome amplification (WTA). cDNA product was purified by 0.65× AmPure XP beads (Beckman Coulter, catalog #: A63882), and Hashtag product was isolated from unbounded fraction by adding additional 0.7× AmPure XP beads (final 1.35×). cDNA and hashtag libraries were then subjected to 2nd round of WTA and 2nd round of hashtag-amplification, respectively, followed by purification by AmPure XP beads. Illumina cDNA libraries were constructed from amplified cDNA libraries by using NEBNext Ultra II FS library prep kit for Illumina (New England Biolabs, catalog #: E7805).

cDNA and barcoded hashtag libraries were sequenced on an Illumina Novaseq 6000 sequencer (Illumina, catalog #: 20012850) with an Illumina Novaseq 6000 S4 Reagent Kit v1.0 (200 cycles) (Illumina, catalog #: 20027466) or NovaSeq 6000 S4 Reagent Kit v1.5 (200 cycles) (read1 67 base-pair and read2 155 base-pair) (Illumina, catalog #: 20028313). Pair-end Fastq files (R1: cell barcode reads, R2: RNA reads) of TAS-Seq and BD WTA kit data were processed as follows. After adapter removal and quality filtering by Cutadapt-2.10[43] in R software 3.6.3, gene expression libraries were aligned to mouse Ensemble RNA by Bowtie2-2.4.2[44], and count matrices were generated using the modified python script of BD Rhapsody workflow. Valid cell barcodes were identified as cell barcodes above the inflection threshold of knee-plot of total read counts of each cell barcode identified by the DropletUtils package[45]. Each sample origin and doublets were identified based on fold-change of the normalized read counts of Hashtag antibodies. To subtract the background read counts of each gene caused by RNA diffusion during the lysis step within the BD Rhapsody cartridge and reverse transcription, distribution-based error correction (DBEC) was performed[42]. The resultant dataset was mainly analyzed using R software package Seurat v4.0.4[46] in R 4.1.0. As quality control, doublets were filtered out. The log-normalized gene counts were calculated using NormalizeData function (scale.factor = 1,000,000) and highly variable genes were defined by FindVariableFeatures function (selection.method = "vst", nfeature = 2000). To mitigate the effect of cell cycle heterogeneity and the difference in read counts, cell cycle scores were calculated using CellCycleScoring function. Read counts and the difference between the G2M and S phase scores were regressed out by the ScaleData function. Principal component analysis was performed on the variable genes, and principal components with their *p* value < 0.05 calculated by the jackstraw method, were subjected to cell clustering (resolution = 0.2 for Figs. 1 and 5: 1.0 for Fig. 2) and UMAP dimensional reduction. For Fig. 2, basophil-lineage clusters were subjected to re-clustering analysis. For the joint analysis of basophils in steady state mice (Fig. 2) and Nb-infected mice (Fig. 5), basophil clusters in these two scRNA-seq datasets were merged by the SelectIntegrationFeatures function, FindIntegrationAnchors function, and IntegrateData function. Differentially expressed genes were defined as those whose *p* value, as calculated by the Wilcoxon rank sum test and adjusted by the Bonferroni method is <0.05 and whose log₂[Fold change] is >0.5 or <−0.5. GSEA and GO

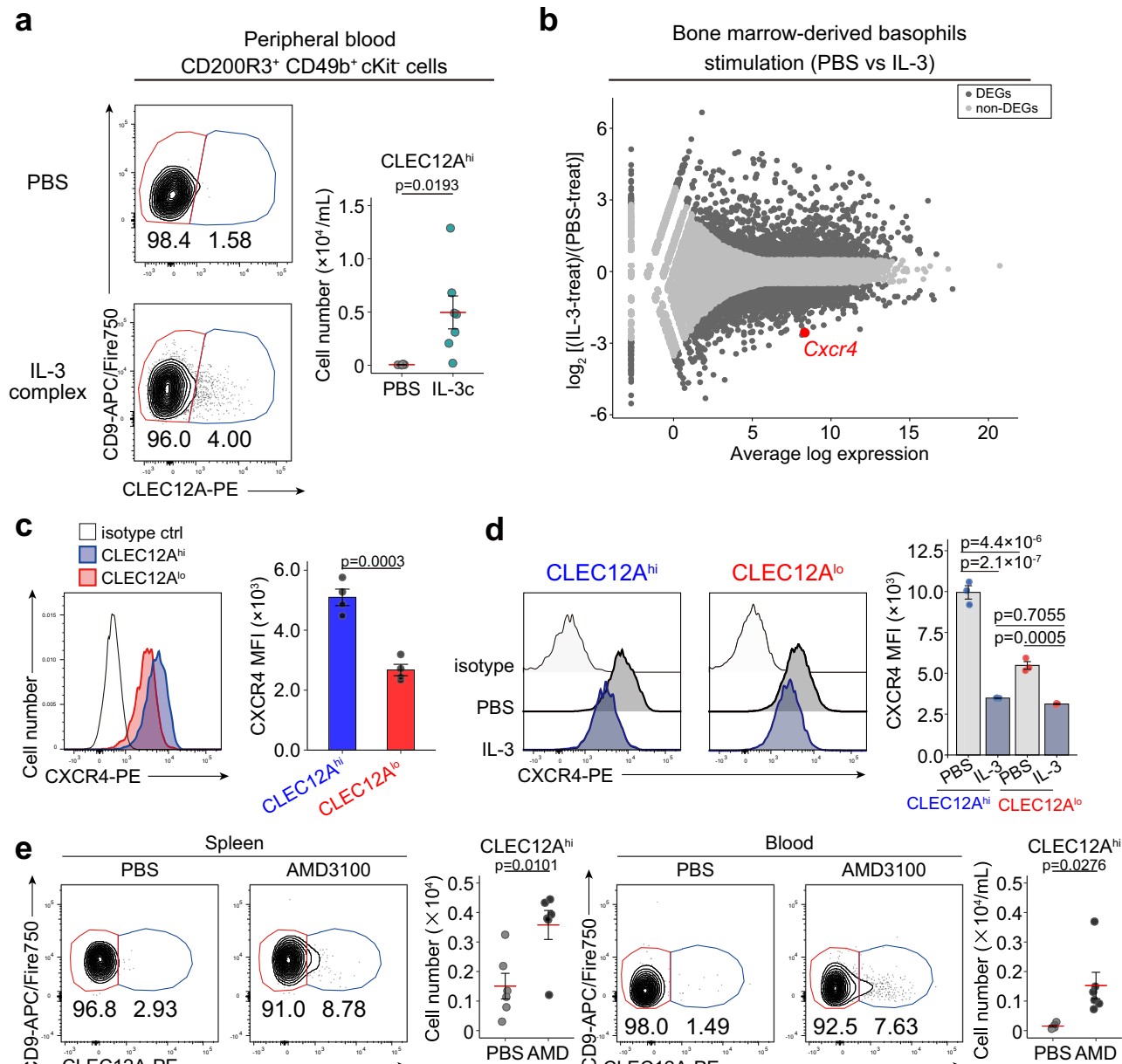

**Fig. 6 | IL-3-mediated CXCR4 down-regulation on pre-basophils promotes their egress from the bone marrow. a** C57BL/6 mice were intraperitoneally administered with IL-3 complex or control PBS. Surface expression of CLEC12A and CD9 in cKit⁻CD200R3⁺CD49b⁺ basophils isolated from the peripheral blood 4 days after the treatment is shown (left panels). The number of CLEC12A^hi basophils in the peripheral blood of the treated mice is shown (right panel, mean ± SEM, *n* = 7 each) (**b**) BMBAs were stimulated with IL-3 or control PBS for 6 h and subjected to bulk RNA-seq analysis. The MA plot of genes expressed in PBS- versus IL-3-treated BMBAs is shown. Black dots indicate differentially expressed genes (DEGs) while gray dots indicate non-differentially expressed genes (non-DEGs). Bulk RNA-seq data were obtained in *n* = 3 cultures for each stimulation. **c** Cell surface expression of CXCR4 in CLEC12A^hiCD9^lo (blue shaded histogram) and CLEC12A^lo CD9^hi (red shaded histogram) basophils isolated from the bone marrow of non-treated mice is shown (left panel). Open histogram indicates control staining with isotype-matched control. MFI of CXCR4 is shown (right panel) (mean ± SEM, *n* = 4 each). **d** CD49b⁺

cells isolated from the bone marrow of non-treated mice were cultured ex vivo with IL-3 (10 ng/mL) or control PBS for 4 h. CXCR4 expression on CLEC12A^hiCD9^lo and CLEC12A^lo CD9^hi basophils after the treatment with IL-3 (blue shaded) or control PBS (gray shaded) is shown (left panel). Open histograms indicate control staining with isotype-matched control. MFI of CXCR4 is shown (right panel) (mean ± SEM, *n* = 3 each). **e** Mice were intraperitoneally treated with AMD3100 (AMD), a CXCR4 inhibitor, or vehicle (PBS) alone. One hour after the treatment, the spleen and peripheral blood cells were separately isolated and subjected to flow cytometric analysis. The surface expression of CLEC12A and CD9 in cKit⁻CD200R3⁺CD49b⁺ basophils and the number of CLEC12A^hiCD9^lo basophil subpopulation are shown (mean ± SEM, *n* = 6 each). Data shown in (**a**) and (**c–e**) are representative of at least three independent experiments. Data in (**b**) were obtained from a single experiment. Unpaired Student's *t* test was used for comparing two groups (**a**, **c**, **e**) and two-way ANOVA with Tukey's multiple comparisons test for multiple comparisons (**d**).

enrichment analysis were conducted by utilizing the R software package clusterProfiler v4.0.5[47]. Pseudotime analysis was performed by utilizing monocle3 v1.0.1[48]. RNA velocity analysis was conducted by utilizing scVelo v0.2.5[49] under python v3.8.15 and R package reticulate v1.28.

**Bulk RNA-seq analysis**

In bulk RNA-seq analysis in Fig. 4a–c, Fig. 6b, and supplementary Fig. 11, total RNA was lysed from 5000 to 80,000 cells of sort-purified basophils (Fig. 4a–c), 10,000 cells of BMBAs (Fig. 6b), or 10,000 cells of stimulated basophils (Supplementary Fig. 11) by using lithium

dodecyl sulfate-based lysis/storage buffer[50,51]. In bulk RNA-seq analysis in Supplementary Fig. 12, total RNA was purified by NucleoSpin RNA Plus XS (Macherey-Nagel, catalog # U0990C). mRNA was isolated from total RNA by Dynabeads M-270 streptavidin (Thermo Fisher Scientific) immobilized with biotin-EcoP-(dT)25 (Integrated DNA Technologies). On-beads reverse transcription was performed by Superscript IV reverse transcriptase (Thermo Fisher Scientific), and free primers were digested by Exonuclease I treatment. Then, polyC-tailing, second-strand synthesis, 1st round of WTA, and second round of WTA was performed by TAS-Seq workflow[42]. In Fig. 4a–c, Fig. 6b, and supplementary Fig. 11, whole-transcriptome library was subjected to fragmentation/end-repair/A-tailing using NEBNext Ultra II FS DNA Library Prep Kit for Illumina (New England Biolabs, catalog #: E7805). Ligated products were purified and performed the barcoding PCR. Pooled libraries were sequenced by Illumina Novaseq 6000 sequencer (Illumina) and NovaSeq 6000 S4 Reagent Kit v1.5 (200 cycles) (Illumina, catalog #: 20028313)[51]. In Supplementary Fig. 12, whole-transcriptome library was subjected to fragmentation/end-repair/A-tailing using NEBNext Ultra II FS DNA Library Prep Kit for Illumina (New England Biolabs, catalog #: E7805), followed by ligation of adapter carrying Ion-Barcode-common sequence 1, the enrichment of the desired cDNA library molecules by PCR, and size selection by AMPure XP beads (Beckman Coulter)[51]. Sequencing was performed by using an Ion Hi-Q Chef kit (Thermo Fisher Scientific, catalog #: A27198), an Ion PI v3 Chip kit (Thermo Fisher Scientific, catalog#: A26770), and an Ion Proton Sequencer (Thermo Fisher Scientific) according to the manufacturer's instructions.

Adapter trimming and quality filtering of sequencing data were performed by using Cutadpat-v3.4. The filtered reads were mapped to reference RNA (GRCm38 release-101) using Bowtie2-2.4.2, and read number of each gene were counted. Normalization of count data and DE analyses were performed by utilizing the R software package TCC v.1.32.0[52]. GO enrichment analysis was performed by the R software package clusterProfiler v4.0.5[47].

### Adoptive transfer of basophils into irradiated mice

Pre-basophils and mature basophils ($5 \times 10^4$ cells) sort-purified from the bone marrow of CD45.2$^+$ C57BL/6 J wild-type mice, together with unsorted CD45.1$^+$ BM cells ($0.3 \times 10^6$ cells), were intravenously injected into 9-Gy X-ray irradiated CD45.1 congenic mice. One day later, the bone marrow cells were isolated and subjected to flow cytometric analysis. In this experiment, CD34$^+$ cells were not excluded from CLEC12A$^{hi}$ basophils.

### Treatment of mice with cytokines or inhibitors

For the treatment of IL-3 complexes, 10 μg of recombinant IL-3 (BioLegend) was mixed with 5 μg of anti-IL-3 antibody (BioLegend; clone: MP2-8F8, catalog#: 503910) to form an IL-3 complex[53]. The IL-3 complex or control PBS was then intraperitoneally administered to mice. Single cell suspensions were prepared from the peripheral blood of mice 4 days after administration and subjected to flow cytometric analysis. For CXCR4 inhibitor treatment, mice were intraperitoneally treated with a single administration of AMD3100 (200 μg per mouse; abcam) or control PBS. Single cell suspensions were prepared from the spleen, and peripheral blood 1 h after treatment and subjected to flow-cytometric analysis.

### Helminth infection

Mice were subcutaneously infected once or twice with third-stage larvae (L3) of *Nippostrongylus brasiliensis* (Nb). For the primary Nb infection model, mice were injected with 500 L3 in the tail base, and the infected lungs, spleen, bone marrow, and peripheral blood were collected at 3, 5 or 7 days-post infection. For the secondary Nb infection model, infected mice were injected with 500 L3 in the flank 18 days

after the first infection. Two days after the secondary Nb infection, the bone marrow and skin at the inoculation site were harvested.

### Ex vivo cell culture of sort-purified basophils

For ex vivo cell culture experiments in Fig. 3a–c, sort-purified pre-basophils and mature basophils ($1 \times 10^4$ cells) were separately cultured for 1 or 2 days in the absence of IL-3, followed by flow cytometric analysis. In Fig. 4e, f, mice were first intravenously administered with 300 μg of TNP-specific IgE antibody (IgE-Lb4) for IgE-sensitization to basophils, 1 day prior to basophil isolation. Sort-purified pre-basophils and mature basophils ($2 \times 10^4$ cells) were incubated with TNP-conjugated ovalbumin (TNP-OVA: 10 ng/mL) or control OVA (10 ng/mL) at 37 °C for 1 h or 6 h. In Fig. 4g, pre-basophils and mature basophils ($2 \times 10^4$ cells) were isolated from the bone marrow of non-sensitized mice and stimulated with the following reagents at 37 °C for 6 h: IL-3 (BioLegend: 10 ng/mL), IL-18 (BioLegend: 10 ng/mL), IL-33 (BioLegend: 10 ng/mL), LPS (Sigma: 100 ng/mL). The concentration of IL-4 in culture supernatants was measured by using ELISA MAX Standard Set Mouse IL-4 (BioLegend, catalog #: 431101).

### Statistical analysis

Statistical analyses were conducted by using GraphPad Prism (ver 7.0.3) or R software (ver 4.1.0). Results were displayed as means ± SEM. Comparisons between two groups were performed using unpaired Student's *t* test. Comparisons among multiple groups were performed using one-way or two-way analysis of variance (ANOVA) with post hoc Tukey's HSD test. A *P* value < 0.05 was considered statistically significant.

### Reporting summary

Further information on research design is available in the Nature Portfolio Reporting Summary linked to this article.

## Data availability

Transcriptomic data generated in this study have been deposited in Gene Expression Omnibus (GEO) under accession numbers GSE206589, GSE206590, GSE206591, GSE206592, GSE206593, and GSE206631. Source data are provided as a Source Data file. Source data are provided with this paper.

## Code availability

The code is available at GitHub (https://github.com/KensukeMiyake/Pre-basophil-paper, https://doi.org/10.5281/zenodo.7769735).

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

## Acknowledgements

We thank Dr. William E. Paul for providing IL-4 reporter G4 (*Il4*^GFP/+) mice; ImmunoGeneTeqs, Inc. for support for scRNA-seq analysis; K. Takahashi for technical support; S. Yoshikawa and all the members of the Karasuyama laboratory for helpful discussions. Cell sorting by using FACS AriaIII and scRNA-seq analysis by using BD Rhapsody system were performed in part at Research Core of Tokyo Medical and Dental University (TMDU). This work was supported by research grants from the Japanese Ministry of Education, Culture, Sports, Science and Technology [20K16277 (K.M.), 22K007115 (K.M.), 19H01025 (H.K.), 22H02845 (H.K.), 22H05064 (S.S.)], Takeda Science Foundation (K.M.), KANAE Foundation for the Promotion of Medical Science (K.M.), The Uehara Memorial Foundation (K.M.), The Naito Foundation (K.M.), and Ohyama Health Foundation (K.M.) JST SPRING Grant Number JPMJSP2120 (J.I.), the Japan Agency for Medical Research and Development PRIME program (Grant Number JP21gm6210025) (S.S.), Young Innovative Medical Scientist Unit by Tokyo Medical and Dental University (K.M.), and TMDU priority research areas grant (K.M.).

## Author contributions

K.M., J.I., J.N., S.S., K.I., and H.K. designed the research. K.M., and J.I., performed most of the experiments and analyzed data. K.I. performed parasite infection experiments. S.S. performed scRNA-seq analysis. K.M., J.I., J.N., and S.S. analyzed transcriptome data. K.M. and H.K. supervised the work. K.M., J.I., and H.K. wrote the paper. All authors provided critical review of the paper.

## Competing interests

S.S. reports advisory role for ImmunoGeneTeqs, Inc; stock for ImmunoGeneTeqs, Inc. The remaining authors declare no competing interests.
