## [Peer Review File · Nature Communications]

Single cell transcriptomics resolves the basophil differentiation trajectory and identifies pre-basophils upstream of mature basophilsREVIEWER COMMENTS

Reviewer #1 (basophil development) (Remarks to the Author):

The study by Miyake has comprehensively characterized basophil differentiation at the single-cell level in mouse hematopoiesis. The single-cell transcriptomics data expands previously available data with a huge number of cells, facilitating the identification of new cell stages in bone marrow and the periphery. This is complemented with novel mechanistic insights into the egress of immature basophils from bone marrow into the blood, and the basophils' migration into the periphery in infection. The data is overall clearly presented, with sufficient numbers of experimental repeats. The reviewer's main concern regards whether immature basophils indeed represent an undiscovered population, or whether it is overlapping with the previously described basophil progenitor population. The reviewer comments follow:

Major comments:

Comment 1: The manuscript states that the basophil differentiation pathway from basophil progenitor to mature basophils remained ill-defined (Abstract line 18-19 and discussion lines 276-277). However, reference 20 (Hamey et al) has performed single-cell transcriptomics analysis of cells spanning basophil progenitors to mature basophils (using Smart-seq2). Hamey et al for example shows that FcεRI downregulation and CD49b upregulation (at the protein level) correlate with basophil differentiation. The same study also shows that Mcpt8 and Prss34 are downregulated (at the gene expression level) with differentiation. Given reference 20, it is surprising that Miyake et al initially hypothesize the opposite, i.e. Clec12a hi basophils (that express high Mcpt8 and Prss34) are more mature-type basophils compared with Clec12a lo basophils (lines 139-144). Hamey et al further demonstrates that cell cycling is associated with the most immature basophils or BaP, whereas cell cycling is lost in the most differentiated basophils, which is one of the key messages in Fig 4 of the present manuscript. Overall, stating that basophil differentiation is ill-defined seems to be an exaggeration.

Comment 2: The gene expression profiles of BaP and immature basophils appear strikingly similar (Fig 4A), and it is puzzling that the BaP are not shown in the single-cell transcriptional data analysis in Fig 2E (as BaP express high levels of Mcpt8). Further evidence that immature basophils represent a unique population (separate from BaP) is therefore warranted. For example, how do the populations (BaP, immature Ba, and mature Ba) compare in flow cytometry analysis and morphologic assessment? How do the single-cell transcriptomics in the present manuscript relate to the previously published BaP and basophil populations described in Hamey et al? Addressing the relationship between the three stages is particularly important to highlight as the present study's main message is that it fills the gap between basophil precursors and mature basophils. Yet, little evidence is presented regarding the relationship between BaP and immature basophils. Showing a comprehensive single-cell dataset with all 3 populations present would be one of the most convincing approaches.

Comment 3: Related to RNA-sequencing datasets: Please specify the number of independent experiments in each figure. If one, please indicate this. Also specify the number of animals used to generate the datasets. Were cells pooled from several animals, are data available from each animal, or was only one animal used for each condition to generate the single-cell RNA-seq data. Does n=3 for bulk RNA-seq data indicate 3 animals/cultures or 3 independent experiments?

Comment 4, related to Fig 5: A sham-treated/naïve, or day 0 group showing the basophil populations in bone marrow, spleen, blood, lung and skin is needed for the data presented in Fig 5A-B. Do "Mature2" in Fig 5C really represent basophils? What about their expression of other basophil-related transcripts such as Hdc and Cebpa? The population appears unique from basophils in uninfected mice (Fig S7).

Minor comments:

The identification of CLEC12A on human peripheral blood basophils should be discussed (previously described by Toft-Petersen et al, DOI: 10.1002/cyto.b.21540)

Line 76: Remove the statement that no cell line equivalent to primary basophils is available. Cell lines with basophil characteristics are available, for example KU812 cells.

Lines 89-93: Provide reference that peripheral blood basophils in mouse display a ring-shaped nuclei. Specify that “this” on line 91 refers to the larger cell body.

Lines 196-199 and related to Fig 4E. The results section hints that high CD63 levels indicate more activation. However, it is also possible that immature basophils have less CD63 in their granules or that they have less granules. Therefore, the study should also show the percentage of activated cells comparing immature and mature basophils.

Lines 201-202: The conclusion that “immature basophils appear to be more reactive to innate stimuli whereas mature basophils are more reactive to adaptive stimuli” is too far-reaching given the limited number of stimuli tested. Rephrase.

Line 214 and Fig S5B: S5B is visually difficult to interpret. There appears to be similarities between the mature basophils in each condition, and immature basophils in each condition. However, the reviewer suggests another way of showing that the conditions generate basophils with similar gene expression profiles. How about PCA analysis?

Line 224: Remove “unexpectedly”

Line 244: Why is a new gene set presented in Fig 5E? It would make more sense to use the gene sets presented in Fig 4B again.

Lines 282-299 are highly speculative and can be discussed in a broader context using a few sentences rather than a large paragraph.

Line 469: Typo: 5G should be 4G.

Line 473: Specify the number of cells that were activated.

Specify cell culture medium

The strain of mice used for each experiment should be specified (BALB/c and/or C57BL/6J are specified in the methods section). Why were some experiments performed using BALB/c and others C57BL/6J?

Proof-reading for typos and grammar would improve the manuscript (for example missing “the” in the title “...the basophil trajectory...”).

Reviewer #2 (basophil biology) (Remarks to the Author):

First the authors perform a titration experiment and determine that 0.3ng/ml of IL-3 is ideal for highly purified BM-derived basophils. In these cultures, they then identify that basophils can be developed into two subpopulations based in CD49b and Fcεr1 expression. These cells also displayed distinct cell morphologies. Next the authors ran single cell RNA sequencing analysis on the cells and we able to identify several subsets of basophils or basophil-like cells based on expression levels of CD49b and Fcεr1. They used these data to identify the surface markers CD9 and Clec12a that were differentially expressed by these clusters. The then show that that these markers can be used by flow to distinguish the basophil clusters. With CD49^{bhi} Fcεr1^{lo} cells staining CD9^{hi} and Clec12^{low} and the opposite for Fcεr1^{hi} Cd49b^{low} cells. They then show that both of these clusters can be found in the BM of mice, but only the CD49^{bhi} Fcεr1⁺ (CD9^{hi} Clec12^{low}) cells can be found in the blood and spleen of mice. The authors then use a Mcpt8GFP mouse model to run single cell RNA sequencing on GFP⁺ cells from the BM and spleen respectively. Upon analysis, they could identify 4 clusters of basophils (Baso1-4) in each compartment with cultures 1 and 2 being much more prevalent in the spleen. Monocle analysis suggested that Clec12a was continuously changed as the basophils with the Celc12a high cells expressing higher levels of Mcpt8 and Fcεr1 and therefore likely to be more mature. The authors then test this by culturing the basophils with each phenotype. They demonstrate that the Fcεr1^{hi} basophils have a longer life span and transition into the Clec12a low population over time. Similar results were observed in vivo. The authors then compare these basophil populations by single cell RNA sequencing along with BaPs. They find that the Celc12a high population resembles

BaPs while the Clec12a low population is more like mature basophils found in the spleen. Consistent with this, immature Clec12a basophils were more proliferative. Interestingly, immature basophils were less responsive to IgE and more responsive to IL-33 as measured by IL-4 production. The authors then perform primary and secondary infections with Nippo and show that immature Clec12a hi basophils are found in the spleen blood and lung (primary) and skin (secondary) post infection. Single cell RNA sequencing analysis of these immature basophils in the skin suggest they are similar to those in the bone marrow. Finally, the authors show that a down regulation of IL-3 by Cxcl4 appears to promote the egress of immature basophils into the periphery.

This is a very interesting study by a well-respected group with great expertise in basophil biology. The topic is of importance and the authors nicely show the existence of a transitional immature basophil population. They have also identified surface markers that allow these cells to be separated. Perhaps most importantly they show indications that these cells are functionally distinct. The similar actions of these cells to TSLP elicited basophils is fascinating and suggests that they may perform unique roles compared to their mature counterparts. However, the manuscript stops short of demonstrating the functional significance to these cells. It's unclear if the authors believe this exists or if they have just identified a transition process occurring during maturation from a BaP to a mature basophil. The short life span and high turnover of basophils makes it unlikely that a very small population of these immature cells are really needed to maintain a presence in the periphery. Why would the cells have distinct functions? Is a certain percent of them needed to respond to IL-33 and IgE to promote optimally IL-4 production. Are they even making cytokines in the periphery? Overall, the manuscript is well done, the data are convincing, but the conceptual advance is not very clear. The authors need to show some functional aspect of this cell that makes it truly distinct from a BaP.

Minor points

Are the authors gating off CD34 for figure 1A? It's showed in other figures, but that should be made clear for Fig1A.

Why did they transition to the MCPT8 model when they clearly show that Mcpt8 is differentially expressed between the 2 clusters identified in the cultures in Figure 1F. It was more differentially expressed than Fcεr1 that they are using to segregate their clusters. Given the difference in expression, can they prove that they were capturing both populations faithfully with this reporter?

The lines stated from 140-144 indicate that they Clec12ahi cells are more mature. In contrast, the authors then describe the Clec12a low cells as more mature (line 159). This was a bit confusing considering that the Clec12a hi cells were more prevalent in the BM and therefore much more likely to be immature.

The flow plots showing changes in CD9 and Clec21a are clear but should be shown as bar graphs with statistics when appropriate. The use of a single flow plot makes it difficult to tell how representative the changes are.

For Figure 5A the authors should show the day 0 graphs as controls. The amount of Clec12ahi basophils appears to be extremely low compared Clec12a low. The authors should more clearly represent the comparison by having them on the same graph.

Reviewer #3 (scRNAseq and data analyses) (Remarks to the Author):

Recent work has identified a CD34+cKit-CD200R3+CLEC12A+ basophil precursor (BaP) that is developmentally restricted to the basophil lineage. The developmental stages downstream of these BaPs remain to be further elucidated. In this body of work the authors leverage scRNAseq, in vitro differentiation assays and adoptive transfers to propose a cKit-CD200R3+CD9hiCLEC12A+ precursor population to be downstream of BaPs. While this study aims to address the identity of a precursor

population downstream of the described BaPs, it remains unclear if the immature basophils (CLEC12A^{hi}CD9^{lo}) described in this body of work is comprised of a homogenous group of precursors or if it is a heterogenous mixture of precursors and mature cells. From a computational standpoint this concern could be addressed in the following ways:

- In Figure 4 the authors leverage bulk RNAseq to compare BaPs to immature CLEC12A^{hi}CD9^{lo} and mature CLEC12A^{lo}CD9^{hi} basophils. While bulk data does not allow dissection of the heterogeneity within these immature basophils, it can be used to identify transcriptionally similar cells in the authors scRNAseq data. This could be achieved by integrating the bulk data into the scRNAseq data or by calculating scores based on gene lists obtained from the bulk RNAseq data.
- In line with the previous remark, a clear absence of cells in-between the Baso3 and Baso2 populations can be observed in the scRNAseq data obtained from the spleen (Figure 2E). The absence of these cells could be interpreted as them being upstream of the cells detected in both the spleen and bone marrow. Therefore, it would be informative to leverage RNA velocity to provide a prediction with regards to the directionality of development. This should allow the authors to either validate the trajectory predicted by monocle or generate a more accurate trajectory.
- These predicted developmental pathways could further be dissected by integrating the scRNAseq data generated in this study with previously published datasets (Dahlin et al, 2018; Weinreb et al 2020). While this might provide insights into the developmental hierarchy of basophil populations described here, it is possible that this approach might not be feasible due to the technical limitations associated with integrating scRNAseq datasets obtained using different technologies.
- In Figure S3 a re-analyzed UMAP is visualized that contains cells derived from both the spleen and bone marrow. It would be informative if the authors would be able to visualize Cd34 and Kit RNA expression as is the case in Figure S3C. Moreover, cell cycle scores visualized on top of the UMAPs would be informative for the reader as this suggests that certain populations are actively proliferating and potentially developing.
- It is unclear as to whether the secretion of IL4 upon stimulation with different cytokines is an inherent function of basophil precursors or can be attributed to residual contaminants within the CLEC12A^{hi}CD9^{lo} population. The authors could leverage ligand-receptor prediction algorithms to predict the receptors and their downstream molecules that are activated by the cytokines leveraged in Figure 4G. These can then be visualized onto the relevant UMAPs to assess potential heterogeneity within the immature basophils. In case these pathways have been thoroughly described, the relevant genes can be directly visualized without the need for ligand-receptor predictions.
- In figure 6 the authors describe the role of CXCR4 down regulation that allows immature basophils to emigrate from the bone marrow. The authors however do not leverage their scRNAseq data obtained from bone marrow and spleen to corroborate this. It would be interesting to see CXCR4 expression visualized on the different UMAPs in the case that CXCR4 expression not only changes upon emigration out of the bone marrow but also differs along development.
- In both Figure 1D and Figure 5C the UMAPs have not been re-analyzed, limiting the interpretability for the reader. It would be interesting to see these populations in more detail after going through re-analysis.
- In figure 5C it is unclear whether the CD9^{lo}CLEC12A^{hi} cells derived from the skin are enriched to equal numbers. Clarifying this would facilitate the interpretation of the figure. In addition, it is unclear whether the labels immature and mature are computationally defined (eg clusters) or if these correspond to the sorting strategy used.
- It is unclear whether cKit-CD200R+CD9^{hi}CLEC12A+CD34+ BpAs were excluded during sort purification prior to the adoptive transfer experiment in Figure 3C. Stating this in the methods, figure or legend would facilitate the interpretation of that figure.

If the authors would be willing to address this concern and adjust their conclusions if needed I would be supportive of this manuscript being published in Nature Communications.

Response to Reviewer 1

“The study by Miyake has comprehensively characterized basophil differentiation at the single-cell level in mouse hematopoiesis. The single-cell transcriptomics data expands previously available data with a huge number of cells, facilitating the identification of new cell stages in bone marrow and the periphery. This complemented with novel mechanistic insights into the egress of immature basophils from bone marrow into the blood, and the basophil’s migration into the periphery in infection. The data is overall clearly presented, with sufficient numbers of experimental repeats.”

We greatly appreciate the reviewer’s favorable comments.

Responding to the Reviewer’s main concern

“The reviewer’s main concern regards whether immature basophils indeed represent an undiscovered population, or whether it is overlapping with the previously described basophil progenitor population.”

We thank the Reviewer for bringing this important issue to our attention. To address the Reviewer’s main concern “whether immature basophils indeed represent an undiscovered population or whether it is overlapping the previously described basophil progenitor population”, we reexamined the possible relationship between basophil progenitors (pre-BMPs and BaPs) and CLEC12A^{hi} immature basophils by reanalyzing our transcriptomic data together with the publicly available scRNA-seq dataset and by performing additional flow cytometric analyses as described in detail below. Based on the obtained results, we concluded that the previously-defined CD34⁺ BaPs constitute a minor fraction (approximately 10%) of the CLEC12A^{hi} basophil population identified in the present study. The scRNA-seq analysis revealed that the *Cd34*⁺ and *Cd34*^{lo} basophils expressing CLEC12A^{hi} cannot be segregated as distinct clusters, and hence the previously-defined BaPs do not seem to represent a particular stage of the basophil development in terms of the gene profiling. Accordingly, we designated CLEC12A^{hi} basophils as pre-basophils in place of immature basophils.

We greatly appreciate the Reviewer’s valuable comments and suggestions, leading to the more precise and clearer view of the differential trajectory of basophils.

Based on the data obtained in additional experiments, we revised our manuscript as follows.

(Abstract: P.2 Line 20-25 in the revised Ms.)

Combined with flow cytometric and functional analyses, **we identified** previously-unappreciated **c-Kit**⁺**CLEC12A**^{hi} **pre-basophils located downstream of pre-basophil and mast cell progenitors (pre-BMPs) and upstream of CLEC12A**^{lo} **mature basophils. The transcriptomic analysis revealed that the previously-defined basophil progenitors constitute a minor fraction of pre-basophils and do not represent a particular stage of the basophil development.**

(Introduction: P.4 Line 69-72 in the revised Ms.)

In the present study, we sought to address this issue by conducting highly sensitive scRNA-seq analysis of basophils in mice and identified previously unappreciated **pre-basophils** located between **pre-BMPs** and mature basophils along the basophil ontogeny. **The previously-defined BaPs turned out to be a minor fraction of pre-basophils in terms of gene profiling.**

(Results: P.6 Line 126-132 in the revised Ms.)

Of note, $\text{Lin}^- \text{cKit}^+ \text{CD34}^+ \text{CD200R3}^+$ BaPs also displayed the $\text{CLEC12A}^{\text{hi}} \text{CD9}^{\text{lo}}$ phenotype (**Supplementary Fig. S3a**), and approximately 10 % of the $\text{CLEC12A}^{\text{hi}}$ population among $\text{cKit}^+ \text{CD200R3}^+$ basophil-lineage cells showed low expression of CD34 (**Fig. 2d and Supplementary Fig S3b**), indicating that $\text{CLEC12A}^{\text{hi}}$ basophils might contain the uni-potent BaP-like population. Supporting this notion, CD34^+ BaPs and $\text{CD34}^- \text{CLEC12A}^{\text{hi}}$ basophils displayed similar morphology and surface expression profiles (**Supplementary Fig S3c-e**).

(Results: P.7 Line 142-159 in the revised Ms.)

Re-clustering of these three clusters identified **1 cluster of $\text{Fcer1a}^+ \text{Kit}^+ \text{Cd34}^+$ pre-BMP-like cells**¹², **1 cluster of $\text{Clec12a}^{\text{hi}}$ basophils (Baso1)** and **2 clusters of $\text{Clec12a}^{\text{lo}}$ basophils (Baso2 and Baso3)** (**Fig. 2e, f and Supplementary Fig. S4e**). Consistent with the flow cytometric analysis data, $\text{Clec12a}^{\text{hi}}$ basophils (Baso1) were abundant in the bone marrow and much fewer in the spleen (**Fig. 2e, f**). **In line with flow cytometric analysis, 21% of $\text{Clec12a}^{\text{hi}}$ Baso1 cells displayed low levels of Cd34 expression (Fig. 2f and Supplementary Fig. S4e).** Of note, Seurat clustering of Baso1 cells revealed that Cd34^+ and Cd34^- populations could not be segregated as distinct clusters (**Supplementary Fig. S5a-c**), suggesting that Cd34^{lo} BaP-like cells showed gene expression profiles similar to those of $\text{Cd34}^- \text{Clec12a}^{\text{hi}}$ basophils.

To validate our findings, we re-analyzed the publicly available scRNA-seq datasets reported by Weinreb et al.²⁵ (GEO accession number: GSE140802) and identified **1 cluster of $\text{Fcer1a}^+ \text{Kit}^+ \text{Cd34}^+$ pre-BMP-like cells, 2 clusters of $\text{Clec12a}^{\text{hi}}$ basophils (Baso1 and Baso2) and 3 clusters of $\text{Clec12a}^{\text{lo}}$ basophils (Baso3, Baso4, and Baso5)** (**Supplementary Fig. S6a-c**). Even though 11.4 % of $\text{Clec12a}^{\text{hi}}$ basophils displayed low levels of Cd34 , Seurat clustering failed to segregate Cd34^+ and Cd34^- populations as distinct clusters (**Supplementary Fig. S6a-c**) in accordance with the analysis of our scRNA-seq data. Therefore, BaPs do not seem to represent a particular stage of the basophil development in terms of gene profiling.

(Results: P.7 Line 162-164 in the revised Ms.)

Pseudo-time trajectory analysis and RNA velocity analysis of our scRNA-seq data inferred the **basophil differentiation trajectory from pre-BMP-like cells to $\text{Clec12a}^{\text{hi}}$ basophils, and then to $\text{Clec12a}^{\text{lo}}$ basophils (Fig. 2g, h and Supplementary Fig. 7).**

(Results: P.8 Line 189-191 in the revised Ms.)

Accordingly, in the following experiments, we designated $\text{CLEC12A}^{\text{hi}} \text{CD9}^{\text{lo}}$ / $\text{Fc}\epsilon\text{RI}\alpha^{\text{hi}} \text{CD49b}^{\text{lo}}$ and $\text{CLEC12A}^{\text{lo}} \text{CD9}^{\text{hi}}$ / $\text{Fc}\epsilon\text{RI}\alpha^{\text{lo}} \text{CD49b}^{\text{hi}}$ basophils as **pre-basophils** and mature basophils, respectively.

(Results: P.9 Line 202-204 in the revised Ms.)

Of note, **CD34⁻ pre-basophils** displayed a gene expression profile similar to that of **CD34⁺ BaPs** (**Fig. 4a**), strengthening our forementioned conclusion that **CD34⁺ BaPs** are included in the **CLEC12A^{hi} pre-basophil population** in terms of gene profiling.

(Discussion: P.13 Line 302-314 in the revised Ms.)

In the present study, we took advantage of this technology combined with flow cytometric and functional analyses and succeeded in the identification of previously-unappreciated **CLEC12A^{hi}CD9^{lo} pre-basophils** that are developmentally located downstream of pre-basophil and mast cell progenitors (pre-BMPs) and upstream of **CLEC12A^{lo}CD9^{hi} mature basophils** in mice. The majority of pre-basophils express little or no expression of CD34 while approximately 10% of them express low levels of CD34 and therefore likely correspond to BaPs. BaPs were identified in 2005 as unipotent basophil precursors that are downstream of GMPs and differentiate to mature basophils¹⁰. They are characterized by their surface expression of CD34, a transmembrane glycoprotein widely utilized as a marker for hematopoietic progenitor cells³². The scRNA-seq analysis in the present study revealed that the **Cd34⁻ and Cd34^{lo} pre-basophils** cannot be segregated as distinct clusters. Therefore, we here propose that BaPs do not represent a particular stage of the basophil development and rather constitute a minor fraction of the newly-identified pre-basophil population in terms of gene profiling.

Comment #1

“The manuscript states that the basophil differentiation pathway from basophil progenitor to mature basophils remained ill-defined (Abstract line 18-19 and discussion lines 276-277). However, reference 20 (Hamey et al) has performed single-cell transcriptomics analysis of cells spanning basophil progenitors to mature basophils (using Smart-seq2). Hamey et al for example shows that FcεRI downregulation and CD49b upregulation (at the protein level) correlate with basophil differentiation. The same study also shows that Mcpt9 and Prss34 are downregulated (at the gene expression level) with differentiation. Given reference 20, it is surprising that Miyaka et al initially hypothesize the opposite, i.e. Clec12a hi basophils (that express high Mcpt8 and Prss23) are more mature-type basophils compared with Clec12a lo basophils (line 139-144). Hamey et al further demonstrates that cell cycling is associated with the most immature basophils or BaP, whereas cell cycling is lost in the most differentiated basophils, which is one of the key messages in Fig. 4 of the present manuscript. Overall, stating that basophil differentiation is ill-defined seems to be an exaggeration.”

According to the Reviewer’s suggestion, we modified the sentence in Abstract as follows.

(Abstract: P.2 Lines 19 in the revised Ms.)

However, their differentiation pathway **remains to be fully elucidated**.

In addition,

In the Discussion section, the sentence “By contrast, the differentiation pathway from basophil progenitors to mature basophils remained ill-defined” was deleted.

Because the Reviewer kindly noted the paper reported by Hamey et al. as a reference, we re-analyzed their scRNA-seq data (GEO accession number: GSE108074) to compare with our data in the present study. According to the protocol described in the paper, Lin⁻Sca-1⁻FcεRI⁺CD34⁺ BaPs and Lin⁻Sca-1⁻FcεRI⁺CD34⁻ basophils were isolated from bone marrow cells and separately subjected to the scRNA-seq analysis. As shown below, the re-analysis of the GSE108074 data revealed that the majority of “CD34⁺ BaPs” express *Kit* in addition to *Cd34*, therefore most likely correspond to *Kit*⁺*Cd34*⁺ pre-BMPs rather than the previously-defined BaPs. Approximately 10 cells among 47 “CD34⁺ BaPs” appear to be *Kit*⁺*Cd34*⁺ authentic BaPs. Therefore, it seems practically difficult to discuss the identity and difference of phenotypes between their “CD34⁺ BaPs” and our pre-basophils (designated as immature basophils in our original manuscript).

Figure for Comment #1. scRNA-seq data of CD34⁻ basophils and CD34⁺ BaPs obtained by Hamey et al. were reanalyzed. (a) UMAP plots of CD34⁻ basophils and CD34⁺ BaPs were shown. (b) Feature plots showing indicated gene expression in CD34⁻ basophils and CD34⁺ BaPs were shown.

Comment #2

“The gene expression profiles of BaP and immature basophils appear strikingly similar (Fig 4A), and it is puzzling that the BaP are not shown in the single-cell transcriptional data analysis in Fig 2E (as BaP express high levels of Mcpt8). Further evidence that immature basophils represent a unique population (separate from BaP) is therefore warranted. For example, how do the populations (BaP, immature Ba, and mature Ba) compare in flow cytometry analysis and morphologic assessment? How do the single-cell transcriptomics in the present manuscript relate to the previously published BaP and basophil populations described in Hamey et al? Addressing the relationship between the three stages is particularly important to highlight as the present study’s main message is that it fills the gap between basophil precursors and mature basophils. Yet, little evidence is presented regarding the relationship between BaP and immature basophils. Showing a comprehensive single-cell dataset with all 3 populations present would be one of the most convincing approaches.”

We greatly appreciate the Reviewer’s valuable comments and suggestions. As described earlier in “Responding to the Reviewer’s main concern”, we performed additional

flow cytometric analyses of basophil progenitors and reanalyzed our transcriptomic data together with the publicly available scRNA-seq dataset. Based on the obtained results, we concluded that the previously-defined CD34⁺ BaPs constitute a minor fraction of the CLEC12A^{hi} basophil population identified in the present study and do not represent a particular stage of the basophil development in terms of the gene profiling. The identity and difference between BaPs reported by Hamey and the basophil subpopulations identified in the present study were already discussed in “Responding to the Reviewer’s main concern” and “Response to Comment #1”.

Comment #3

“Related to RNA-sequencing datasets: Please specify the number of independent experiments in each figure. If one, please indicate this. Also specify the number of animals used to generate the datasets. Were cells pooled from several animals, are data available from each animal, or was only one animal used for each condition to generate the single-cell RNA-seq data. Does n=3 for bulk RNA-seq data indicate 3 animals/cultures or 3 independent experiments?”

We thank the Reviewer for bringing this important issue to our attention. According to the reviewer’s suggestions, we added information regarding the number of independent experiments in Figure legends as follows:

(Figure Legends: P.28 Line 678 in revised Ms.)

Data in d-f were obtained from a single experiment.

(Figure Legends: P.30 Line 697-698 in revised Ms.)

Data in e-i were obtained from a single experiment.

(Figure Legends: P.33 Line 742-743 in revised Ms.)

Data in a-c were obtained from a single experiment. Those shown in d-g are representative of at least three independent experiments.

(Figure Legends: P.35 Line 764-765 in revised Ms.)

Data in e-g were obtained from a single experiment.

(Figure Legends: P.37 Line 789-790 in revised Ms.)

Data in b were obtained from a single experiment.

We further added detailed information about RNA-sequencing datasets in Materials and Methods section and Figure legends as follows:

(Methods: P.17 Line 422-423 in revised Ms.)

BMBAs generated from 5 mice were pooled to generate scRNA-seq data.

(Methods: P.18 Line 428 in revised Ms.)

Basophils isolated from 5 mice were pooled to generate scRNA-seq data.

(Methods: P.18 Line 432-433 in revised Ms.)

Basophils isolated from 5 mice were pooled to generate scRNA-seq data.

(Figure Legends: P.32 Line 724 in revised Ms.)

(n=3, pooled from 4 animals for each replicate)

(Figure Legends: P.36 Line 775-776 in revised Ms.)

Bulk RNA-seq data were obtained in n=3 cultures for each stimulation.

Comment #4, related to Fig 5:

“A sham-treated/naïve, or day 0 group showing the basophil populations in bone marrow, spleen, blood, lung and skin is needed for the data presented in Fig 5A-B.”

According to the Reviewer’s kind suggestion, mice were infected once with Nb larvae or treated with intradermal injection of PBS as a control of Nb-infection. As shown in **Supplementary Fig. S12a-b** in the revised Ms., CLEC12A^{hi} pre-basophils were rarely detected in the lungs of PBS-treated mice on day 7 post-infection in contrast to the case of Nb-infected mice. Regarding to the basophil infiltration to the skin, we previously showed that basophils were hardly detected in the skin on day 0 of the second Nb infection (Obata-Ninomiya et al. *J Exp Med* 2013). Therefore, we cannot display the surface expression of CLEC12A and CD9 in skin-infiltrating basophils on day 0. The expression of CLEC12A and CD9 on basophils isolated from the bone marrow, spleen and peripheral blood of uninfected mice is shown in Fig. 2a.

Accordingly, we added **Supplementary Fig. S12a-b** to the revised manuscript and modified Results section as follows:

(Results: P.10 Line 248-251 in the revised Ms.)

On day 7 post Nb-infection, we detected CLEC12A^{hi} **pre-basophils**, in addition to CLEC12A^{lo} mature basophils, in the spleen, blood and even lungs (**Fig. 5a**), in contrast to **the observation that pre-basophils largely reside within the bone marrow in uninfected or PBS-injected control mice (Fig. 2a and Supplementary Fig. 12a-b).**

“Do “Mature2” in Fig 5C really represent basophils? What about their expression of other basophil-related transcripts such as *Hdc* and *Cebpa*? The population appears unique from basophils in uninfected mice (Fig S7).”

We thank the Reviewer for bringing this issue to our attention. To answer Reviewer’s inquiry, we assessed the expression of basophil marker genes in the Mature2 basophil cluster. As shown below, Mature2 basophils express *Mcpt8*, *Cd200r3* and *Hdc* at the levels almost comparable to Mature1 basophils and pre-basophils even though Mature2 basophils display lower expression of *Cebpa*. Therefore, Mature2 cells appear to be basophil-lineage cells.

Figure for comment #4. The scRNA-seq dataset presented in Figure 5c were analyzed. Feature plots of basophil clusters (clusters 0,1, and 2 depicted in A) showing the expression of indicated genes.

Minor comment #1:

“The identification of CLEC12A on human peripheral blood basophils should be discussed (previously described by Toft-Petersen et al, DOI: 10.1002/cyto.b.21540)”

According to the Reviewer’s kind suggestion, we added a following sentence to the end of Discussion as follows:

(Discussion: PP. 14-15 Line 360-362 in the revised Ms.)

Human basophils have been shown to express CLEC12A⁴⁰ and therefore further studies are warranted to identify the human counterpart of mouse pre-basophils.

Minor comment #2:

“Line 76: Remove the statement that no cell line equivalent to primary basophils is available. Cell lines with basophil characteristics are available, for example KU812 cells.”

According to the Reviewer’s suggestion, we removed the statement, even though we are not sure that KU812 cells really display functional properties equivalent to those of primary basophils, considering that they are derived from Ph1⁺ chronic myelocytic leukemia.

Minor comment #3:

“Lines 89-93: Provide reference that peripheral blood basophils in mouse display a ring-shaped nuclei. Specify that “this” on line 91 refers to the larger cell body.”

According to the Reviewer’s suggestions, we modified our Ms. as follows:

(Results: P.5 Line 89-90 in the revised Ms.)

Notably, FcεRIα^{lo}CD49b^{hi} BMBAAs displayed ring-shaped nuclei as observed in peripheral blood basophils²⁴ whereas...

(References: P.23 Line 576-578 in the revised Ms.)

24. Mukai et al. Basophils play a critical role in the development of IgE-mediated chronic allergic inflammation independently of T cells and mast cells. *Immunity*. **23**(2):191-202. (2005)

(Results: P.5 Line 92-94 in the revised Ms.)

In accordance with their large cell body, FcεRIα^{hi}CD49b^{lo} BMBAAs displayed higher forward scatter (FSC) compared to FcεRIα^{lo}CD49b^{hi} BMBAAs in flow cytometric analysis (**Fig. 1c**).

Minor comment #4:

“Lines 196-199 and related to Fig 4E. The results section hints that high CD63 levels indicate more activation. However, it is also possible that immature basophils have less CD63 in their granules or that they have less granules. Therefore, the study should also show the percentage of activated cells comparing immature and mature basophils.”

According to the Reviewer’s suggestion, we added bar graphs showing the frequency of CD63^{hi} basophils to Fig. 4e.

Minor comment #5:

“Lines 201-202: The conclusion that “immature basophils appear to be more reactive to innate stimuli whereas mature basophils are more reactive to adaptive stimuli” is too far-reaching given the limited number of stimuli tested. Rephrase.”

According to Reviewer’s suggestions, we rephrased the sentences in Abstract and Results section as follows:

(Abstract: P. 2, Line 25-26 in the revised Ms.)

Pre-basophils were highly proliferative and responded better to **non-IgE stimuli but less to IgE-mediated one than did mature basophils.**

(Results: P. 9, Line 225-226 in the revised Ms.)

Thus, pre-basophils appeared to be more reactive to **non-IgE stimulation** whereas mature basophils are more reactive to **IgE/allergen stimulation.**

Minor comment #6:

“Line 214 and Fig S5B: S5B is visually difficult to interpret. There appears to be similarities between the mature basophils in each condition, and immature basophils in each condition. However, the reviewer suggests another way of showing that the conditions generate basophils with similar gene expression profiles. How about PCA analysis?”

According to the Reviewer’s suggestion, we re-analyzed the bulk RNA-seq data shown in **Supplementary Fig. S11b** (Fig. S5B in the original manuscript) and visualized the data by principal component analysis (PCA) in that PC1 explains the maturation of basophils while PC2 explains the stimulation with IL-3 or TSLP, as shown below. This PCA plot together with the histograms shown in Supplementary Fig. S11b indicate that TSLP-elicited mature basophils display gene expression profiles distinct from those of IL-3-elicited pre-basophils.

We are not sure whether this PCA analysis is better than the hierarchically clustered heatmap of DEGs in order to visually display the similarity and difference among four groups of cells in the present case.

Figure for minor comment #6. The principal component analysis plot for bulk RNA-seq dataset shown in Supplementary Fig. S11b.

Minor comment #7:

“Line 224: Remove “unexpectedly””

According to Reviewer’s suggestions, we removed the word “unexpectedly”.

Minor comment #8:

“Line 244: Why is a new gene set presented in Fig 5E? It would make more sense to use the gene sets presented in Fig 4B again.”

The GO term “chromosome segregation” is one of the top 10 enriched GO terms shown in Fig. 4b. According to the Reviewer’s suggestion, we showed the top 10 enriched GO terms in Supplementary Fig. S15 of the revised Ms.

Minor comment #9:

“Lines 282-299 are highly speculative and can be discussed in a broader context using a few sentences rather than a large paragraph.”

According to the Reviewer’s suggestion, we made that part more concise as follows.

(Discussion: P. 14, Line 332-338 in revised Ms.)

Interestingly, Rodriguez Gomez et al. reported the emergence of FcεRI⁺⁺ basophils in the spleen of colitis model mice where FcεRI⁺⁺ basophils produced significantly higher amounts of IL-4 and IL-6 in response to IL-3 than did FcεRI⁺ basophils³⁶. These characteristics of FcεRI⁺⁺ basophils are consistent with those of pre-basophils identified in the present study. Basophil-derived IL-4 and IL-6 were shown to dampen inflammation in the colon³⁶, suggesting the possible contribution of FcεRI⁺⁺ pre-basophils to the regulation of colitis.

Minor comment #10:

“Line 469: Typo: 5G should be 4G.”

We corrected the typo as follows:

(Methods: P. 20, Line 505-506 in revised Ms.)

In **Fig 4g**, pre-basophils and mature basophils (2×10^4 cells) were isolated from the bone marrow of non-sensitized mice...

Minor comment #11:

“Line 473: Specify the number of cells that were activated. Specify cell culture medium.”

According to the reviewer’s suggestions, we added detailed information on cell culture methods in Methods section as follows:

(Methods: PP. 16-17, Line 392-395 in revised Ms.)

Cell culture was performed in complete RPMI (RPMI 1640 medium (Nacalai tesque) supplemented with 10% FBS (Sigma), 2mM L-glutamine (Nacalai tesque), 1mM sodium pyruvate (Nacalai tesque), 0.055mM 2-mercaptoethanol (Gibco), 100U/mL penicillin/streptomycin (Nacalai tesque), and 0.1mM MEM non-essential amino acids solution (Nacalai tesque)).

(Methods: P.20, Line 499-500 in revised Ms.)

For *ex vivo* cell culture experiments in Fig. 3a-b, sort-purified **pre-basophils** and mature basophils (1×10^4 cells) were separately cultured for 1 or 2 days ...

(Methods: P.20, Line 503 in revised Ms.)

Sort-purified **pre-basophils** and mature basophils (2×10^4 cells) ...

(Methods: P.20, Line 505 in revised Ms.)

In Fig 4g, **pre-basophils** and mature basophils (2×10^4 cells) were isolated ...

Minor comment #12:

“The strain of mice used for each experiment should be specified (BALB/c and/or C57BL/6J are specified in the methods section). Why were some experiments performed using BALB/c and others C57BL/6J?”

Responding to the reviewer’s comments, we added the data using C57BL/6 in Supplementary Fig. 1e. More detailed strain information was also added to Materials and Methods as follows:

(Results: P5. Line 86-88 in revised Ms.)

The frequency of the former subpopulation decreased as the concentration of IL-3 increased **regardless of mouse strains used, either C57BL/6 or BALB/c mice (Supplementary Fig. S1c-e)**

(Methods: P16. Line 367-368 in revised Ms.)

In most of the experiments except for Fig. S1, C57BL/6J background mice were used. In Fig. S1a-d, BALB/c mice were used.

Minor comment #13:

“Proof-reading for typos and grammar would improve the manuscript (for example missing “the” in the title “...the basophil trajectory...”).

Even though the proof-reading by native English speakers was already conducted prior to the submission of the original manuscript, we carefully edited the revised Ms. before the re-submission.

Response to Reviewer 2

“This is a very interesting study by a well-respected group with great expertise in basophil biology. The topic is of importance and the authors nicely show the existence of a transitional immature basophil population. They have also identified surface markers that allow these cells to be separated. Perhaps most importantly they show indications that they cells are functionally distinct. The similar actions of these cells to TSLP elicited basophils is fascinating and suggests that they may perform unique roles compared to their mature counterparts.”

We greatly appreciate the reviewer’s favorable comments.

We revised our manuscript according to the three Reviewers’ valuable comments and suggestions. Please note that “immature basophils” used in the original Ms. is now renamed “pre-basophils” in the revised Ms., because our additional experiments indicated that the previously-defined CD34⁺ BaPs constitute a minor fraction (approximately 10%) of the CLEC12A^{hi} basophil population identified in the present study. The scRNA-seq analysis revealed that the *Cd34*⁺ and *Cd34*^{lo} basophils expressing CLEC12A^{hi} cannot be segregated as distinct clusters, and hence the previously-defined BaPs do not seem to represent a particular stage of the basophil development in terms of the gene profiling. Thus, we identified previously-unappreciated cKit⁻CLEC12A^{hi} pre-basophils that are located downstream of cKit⁺ pre-basophil and mast cell progenitors (pre-BMPs) and upstream of CLEC12A^{lo} basophils along the basophil ontogeny.

Based on the data obtained in additional experiments, we revised our manuscript as follows.

(Abstract: P.2 Line 20-25 in the revised Ms.)

Combined with flow cytometric and functional analyses, **we identified** previously-unappreciated **c-Kit⁻CLEC12A^{hi} pre-basophils located downstream of pre-basophil and mast cell progenitors (pre-BMPs) and upstream of CLEC12A^{lo} mature basophils. The transcriptomic analysis revealed that the previously-defined basophil progenitors constitute a minor fraction of pre-basophils and do not represent a particular stage of the basophil development.**

(Introduction: P.4 Line 69-72 in the revised Ms.)

In the present study, we sought to address this issue by conducting highly sensitive scRNA-seq analysis of basophils in mice and identified previously unappreciated **pre-basophils** located between **pre-BMPs** and mature basophils along the basophil ontogeny. **The previously-defined BaPs turned out to be a minor fraction of pre-basophils in terms of gene profiling.**

(Results: P.6 Line 126-132 in the revised Ms.)

Of note, Lin⁻cKit⁺CD34⁺CD200R3⁺ BaPs also displayed the CLEC12A^{hi}CD9^{lo} phenotype (Supplementary Fig. S3a), and approximately 10 % of the CLEC12A^{hi} population among cKit⁺CD200R3⁺ basophil-lineage cells showed low expression of CD34 (Fig. 2d and Supplementary Fig S3b), indicating that CLEC12A^{hi} basophils might contain the uni-potent BaP-like population. Supporting this notion, CD34⁺ BaPs and CD34⁻ CLEC12A^{hi} basophils displayed similar morphology and surface expression profiles (Supplementary Fig S3c-e).

(Results: P.7 Line 142-159 in the revised Ms.)

Re-clustering of these three clusters identified 1 cluster of *Fcer1a*⁺*Kit*⁺*Cd34*⁺ pre-BMP-like cells¹², 1 cluster of *Clec12a*^{hi} basophils (Baso1) and 2 clusters of *Clec12a*^{lo} basophils (Baso2 and Baso3) (Fig. 2e, f and Supplementary Fig. S4e). Consistent with the flow cytometric analysis data, *Clec12a*^{hi} basophils (Baso1) were abundant in the bone marrow and much fewer in the spleen (Fig. 2e, f). In line with flow cytometric analysis, 21% of *Clec12a*^{hi} Baso1 cells displayed low levels of *Cd34* expression (Fig. 2f and Supplementary Fig. S4e). Of note, Seurat clustering of Baso1 cells revealed that *Cd34*⁺ and *Cd34*⁻ populations could not be segregated as distinct clusters (Supplementary Fig. S5a-c), suggesting that *Cd34*^{lo} BaP-like cells showed gene expression profiles similar to those of *Cd34**Clec12a*^{hi} basophils.

To validate our findings, we re-analyzed the publicly available scRNA-seq datasets reported by Weinreb et al.²⁵ (GEO accession number: GSE140802) and identified 1 cluster of *Fcer1a*⁺*Kit*⁺*Cd34*⁺ pre-BMP-like cells, 2 clusters of *Clec12a*^{hi} basophils (Baso1 and Baso2) and 3 clusters of *Clec12a*^{lo} basophils (Baso3, Baso4, and Baso5) (Supplementary Fig. S6a-b). Even though 11.4 % of *Clec12a*^{hi} basophils displayed low levels of *Cd34*, Seurat clustering failed to segregate *Cd34*⁺ and *Cd34*⁻ populations as distinct clusters (Supplementary Fig. S6a-b) in accordance with the analysis of our scRNA-seq data. Therefore, BaPs do not seem to represent a particular stage of the basophil development in terms of gene profiling.

(Results: P.7 Line 162-164 in the revised Ms.)

Pseudo-time trajectory analysis and RNA velocity analysis of our scRNA-seq data inferred the basophil differentiation trajectory from pre-BMP-like cells to *Clec12a*^{hi} basophils, and then to *Clec12a*^{lo} basophils (Fig. 2g, h and Supplementary Fig. 7).

(Results: P.8 Line 189-191 in the revised Ms.)

Accordingly, in the following experiments, we designated CLEC12A^{hi}CD9^{lo} / FcεRIα^{hi}CD49b^{lo} and CLEC12A^{lo}CD9^{hi} / FcεRIα^{lo}CD49b^{hi} basophils as pre-basophils and mature basophils, respectively.

(Results: P.9 Line 202-204 in the revised Ms.)

Of note, CD34⁻ pre-basophils displayed a gene expression profile similar to that of CD34⁺ BaPs (Fig. 4a), strengthening our forementioned conclusion that CD34⁺ BaPs are included in the CLEC12A^{hi} pre-basophil population in terms of gene profiling.

(Discussion: P.13 Line 302-314 in the revised Ms.)

In the present study, we took advantage of this technology combined with flow cytometric and functional analyses and succeeded in the identification of previously-unappreciated CLEC12A^{hi}CD9^{lo} pre-basophils that are developmentally located downstream of pre-basophil and mast cell progenitors (pre-BMPs) and upstream of CLEC12A^{lo}CD9^{hi} mature basophils in mice. The majority of pre-basophils express little or no expression of CD34 while approximately 10% of them express low levels of CD34 and therefore likely correspond to BaPs. BaPs were identified in 2005 as unipotent basophil precursors that are downstream of GMPs and differentiate to mature basophils¹⁰. They are characterized by their surface expression of CD34, a transmembrane glycoprotein widely utilized as a marker for

hematopoietic progenitor cells³². The scRNA-seq analysis in the present study revealed that the *Cd34*⁺ and *Cd34*^{lo} pre-basophils cannot be segregated as distinct clusters. Therefore, we here propose that BaPs do not represent a particular stage of the basophil development and rather constitute a minor fraction of the newly-identified pre-basophil population in terms of gene profiling.

Responding to The Reviewer's main concern

“However, the manuscript stops short of demonstrating the functional significance to these cells. It's unclear is the authors believe this exists or if they have just identified a transition process occurring during maturation from a BaP to a mature basophil. The short life span and high turnover of basophils makes it unlikely that a very small population of these immature cells are really needed to maintain a presence in the periphery. Why would the cells have distinct functions? Is a certain percent of them need to respond to IL-33 and IgE to promote optimally IL-4 production. Are they even making cytokines in the periphery? Overall, the manuscript is well done, the data are convincing, but the conceptual advance is not very clear. The authors need to show some functional aspect of this cell that makes it truly distinct from a BaP.”

We thank the Reviewer for highlighting these important points. To address the Reviewer's concerns, we assessed the functional aspects of CLEC12A^{hi} basophils accumulating in Nb-infected lungs and demonstrated that they are highly proliferative (**Fig. 5b** in the revised manuscript) and produce IL-4 at the levels comparable to mature basophils in IL-4 reporter G4 mice (**Fig. 5c** in the revised manuscript). Similarly, CLEC12A^{hi} pre-basophils in the infected skin showed *Il4* expression at the levels comparable to CLEC12A^{lo} mature basophils during the second Nb infection as shown in below.

Figure for comment #1. C57BL/6 mice were infected twice with Nb larvae, and CLEC12A^{hi} pre-basophils and CLEC12A^{lo} mature basophils were isolated from the bone marrow and the 2nd infected skin on day2 of the second infection. The expression of *Il4* is shown (mean \pm SEM, n=3 each).

Moreover, we conducted bulk RNA-seq analysis of pre-basophils and mature basophils stimulated with IL-3 or antigen/IgE. IL-3-stimulated pre-basophils displayed gene expression profiles distinct from those of IL-3- or antigen/IgE-stimulated mature basophils (**Supplementary Fig. S10**). IL-3-stimulated pre-basophils display higher expression of *Il13*

and *I110* compared to IL-3- or IgE/antigen-stimulated mature basophils. Considering that IL-10 and IL-13 dampen tissue damage in the lungs during Nb infection (newly added references 34 and 35), pre-basophils may play regulatory roles in Nb-infected tissues.

The details of the experimental data and our interpretation are described in the revised Ms. as follows:

(Results: P.10 Line 251-256 in the revised Ms.)

Pre-basophils isolated from Nb-infected lungs displayed higher *ex vivo* EdU incorporation than did mature basophils (**Fig. 5b**), suggesting that pre-basophils retain their high proliferative capacity even outside of the bone marrow. Moreover, pre-basophils accumulating in the infected lungs showed IL-4 expression comparable to that of mature basophils, implying their possible contribution to Th2 immunity against Nb infection (**Fig. 5c**).

(Results: P.9 Line 222-225 in the revised Ms.)

Furthermore, the bulk RNA-seq analysis revealed that IL-3-stimulated pre-basophils display gene expression profiles distinct from those of IL-3- or antigen/IgE-stimulated mature basophils, including upregulated expression of *I110* and *I113* in the former (**Supplementary Fig. S10**).

(Discussion: PP. 13-14 Line 315-332 in the revised Ms.)

We found that pre-basophils in the bone marrow are highly proliferative in contrast to mature basophils. This is also the case in pre-basophils accumulating in Nb-infected peripheral tissues. Even though the frequency of pre-basophils is much less than that of mature ones in the infected tissues, we assume that their proliferation and differentiation to mature basophils as well as their continuous accumulation in the infected tissues enable the efficient expansion of basophils for the protection against parasitic infection, as in the case of bacterial infections that induce the emergence of immature neutrophils in peripheral tissues³³. Besides the proliferative capacity, pre-basophils were found to display immune responsiveness distinct from that of mature basophils. As expected from their location at the earlier differentiation stage, pre-basophils were less activated in response to IgE-mediated stimulation than did mature ones, even though they express higher levels of FcεRI on the cell surface. Intriguingly, however, pre-basophils responded more vigorously than mature ones to IgE-independent stimulation such as IL-3 and IL-33 in terms of IL-4 secretion. Of note, pre-basophils accumulating in the Nb-infected lungs showed IL-4 expression comparable to that of mature basophils, suggesting their possible contribution to Th2 immunity against Nb infection. The transcriptomic analysis in the present study revealed that IL-3-stimulated pre-basophils display higher expression of *I113* and *I110* compared to mature basophils. Considering that IL-10 and IL-13 dampen tissue damage in the lungs during Nb infection^{34,35}, pre-basophils may play regulatory roles in Nb-infected tissues.

Minor comment #1:

“Are the authors gating off CD34 for figure 1A? It’s showed in other figures, but that should be made clear for Fig1A.”

CD34⁺ cells were not gated off in the experiment shown in Fig. 1a. Even when CD34⁺

cells were excluded, the expression profile of FcεRIα and CD49b on CD200R3⁺c-Kit⁻ BMBAs was comparable to the picture shown in **Fig. 1a** (as shown below), indicating that BMBAs are composed of these two subpopulations regardless of whether CD34⁺ populations are excluded or not.

Figure for minor comment #1. Bone marrow cells were cultured in the presence of 0.3 ng/mL of IL-3 for 7 days. CD34⁻CD200R3⁺cKit⁻ BMBAs were gated (left and middle panel) and the surface expression of FcεRIα and CD49b is shown (right panel).

In the revised Ms, when CD34⁺ cells were excluded, we described in figure legends that CD200R3⁺cKit⁻CD34⁻ cells were isolated (gated). When CD34⁺ cells were not excluded, we described that CD200R3⁺cKit⁻ cells were isolated (gated).

Minor comment #2:

“Why did they transition to the MCPT8 model when they clearly show that Mcpt8 is differentially expressed between the 2 clusters identified in the cultures in Figure 1F. It was more differentially expressed than Fcer1 that they are using to segregate their clusters. Given the difference in expression, can they prove that they were capturing both populations faithfully with this reporter?”

We thank the Reviewer for bringing this important issue to our attention. To avoid the possible unwanted effect of antibody binding, Lin⁻GFP⁺ cells were isolated from the bone marrow and spleen of *Mcpt8*^{GFP} basophil-reporter mice. To address the Reviewer’s concern, we analyzed the GFP expression in mature and pre-basophils (**Supplementary Fig. S4a** in the revised Ms.). The majority of both CLEC12A^{lo} and CLEC12A^{hi} basophils expressed GFP at the comparable levels. Therefore, we assume that both basophil populations were faithfully isolated from *Mcpt8*^{GFP} basophil-reporter mice.

Accordingly, we added the following sentence to the revised Ms.
(Results: P.7 Line 137-139 in the revised Ms.)

We confirmed that the majority of both CLEC12A^{hi} and CLEC12A^{lo} basophil subpopulations expressed GFP (Supplementary Fig. 4a).

Minor comment #3:

“The lines stated from 140-144 indicate that they Clec12ahi cells are more mature. In contrast, the authors then describe the Clec12a low cells as more mature (line 159). This was a bit confusing considering that the Clec12a hi cells were more prevalent in the BM and therefore much more likely to be immature.”

We agree to the Reviewer's comment and therefore deleted the sentence "*Clec12a*^{hi} basophils may represent more mature-type basophils" in the revised Ms.

Minor comment #4:

"The flow plots showing changes in CD9 and Clec21a are clear but should be shown as bar graphs with statistics when appropriate. The use of a single flow plot makes it difficult to tell how representative the changes are."

According to the Reviewer's suggestion, we added bar graphs with statistics showing the frequency of CLEC12A^{hi} and CLEC12A^{lo} basophils to **Fig. 3b and d** in the revised manuscript.

Minor comment #5:

"For Figure 5A the authors should show the day 0 graphs as controls."

According to the Reviewer's suggestion, mice were infected once with Nb larvae or treated with intradermal injection of PBS as a control of Nb-infection. As shown in **Supplementary Fig. S12a-b** in the revised manuscript, CLEC12A^{hi} pre-basophils were rarely detected in the lungs of PBS-treated mice on day 7 post-infection in contrast to those of Nb-infected mice. Regarding to the basophil infiltration to the infected skin, we previously showed that basophils were hardly detected in the skin on day 0 of the second Nb infection (Obata-Ninomiya et al. *J Exp Med* 2013). Therefore, we cannot display the surface expression of CLEC12A and CD9 in skin-infiltrating basophils on day 0. The expression of CLEC12A and CD9 on basophils isolated from the bone marrow, spleen and peripheral blood of uninfected mice is shown in Fig. 2a.

Accordingly, we added Supplementary Fig. S12a-b to the revised manuscript and modified Results section as follows:

(Results: P.10 Line 248-251 in the revised Ms.)

On day 7 post Nb-infection, we detected CLEC12A^{hi} **pre-basophils**, in addition to CLEC12A^{lo} mature basophils, in the spleen, blood and even lungs (**Fig. 5a**), in contrast to **the observation that pre-basophils largely reside within the bone marrow in uninfected or PBS-injected control mice (Fig. 2a and Supplementary Fig. 12a-b)**.

"The amount of Clec12ahi basophils appears to be extremely low compared Clec12a low. The authors should more clearly represent the comparison by having them on the same graph."

According to the Reviewer's suggestion, the time course of cell numbers of both CLEC12A^{hi} pre-basophils and CLEC12A^{lo} basophils in each experimental group is shown in Fig. 5a of the revised Ms.

Response to Reviewer 3

“Recent work has identified a CD34+cKit-CD200R3+CLEC12A+ basophil precursor (BaP) that is developmentally restricted to the basophil lineage. The developmental stages downstream of these BaPs remain to be further elucidated. In this body of work the authors leverage scRNAseq, in vitro differentiation assays and adoptive transfers to propose a cKit-CD200R3+CD9hiCLEC12A+ precursor population to be downstream of BaPs. While this study aims to address the identity of a precursor population downstream of the described BaPs, it remains unclear if the immature basophils (CLEC12AhiCD9lo) described in this body of work is comprised of a homogenous group of precursors or if it is a heterogenous mixture of precursors and mature cells.”

We greatly appreciate the reviewer’s generous and precise advice to improve our manuscript. We revised our manuscript according to the three Reviewers’ valuable comments and suggestions. Please note that “immature basophils” used in the original Ms. is now renamed “pre-basophils” in the revised Ms., because our additional experiments indicated that the previously-defined CD34⁺ BaPs constitute a minor fraction (approximately 10%) of the CLEC12A^{hi} basophil population identified in the present study. The scRNA-seq analysis revealed that the *Cd34*⁺ and *Cd34*^{lo} basophils expressing CLEC12A^{hi} cannot be segregated as distinct clusters, and hence the previously-defined BaPs do not seem to represent a particular stage of the basophil development in terms of the gene profiling. Thus, we identified previously-unappreciated cKit⁺CLEC12A^{hi} pre-basophils that are located downstream of cKit⁺ pre-basophil and mast cell progenitors (pre-BMPs) and upstream of CLEC12A^{lo} basophils along the basophil ontogeny.

Based on the data obtained in additional experiments, we revised our manuscript as follows.

(Abstract: P.2 Line 20-25 in the revised Ms.)

Combined with flow cytometric and functional analyses, we identified previously-unappreciated c-Kit⁺CLEC12A^{hi} pre-basophils located downstream of pre-basophil and mast cell progenitors (pre-BMPs) and upstream of CLEC12A^{lo} mature basophils. The transcriptomic analysis revealed that the previously-defined basophil progenitors constitute a minor fraction of pre-basophils and do not represent a particular stage of the basophil development.

(Introduction: P.4 Line 69-72 in the revised Ms.)

In the present study, we sought to address this issue by conducting highly sensitive scRNA-seq analysis of basophils in mice and identified previously unappreciated pre-basophils located between pre-BMPs and mature basophils along the basophil ontogeny. The previously-defined BaPs turned out to be a minor fraction of pre-basophils in terms of gene profiling.

(Results: P.6 Line 126-132 in the revised Ms.)

Of note, Lin⁻cKit⁺CD34⁺CD200R3⁺ BaPs also displayed the CLEC12A^{hi}CD9^{lo} phenotype (Supplementary Fig. S3a), and approximately 10 % of the CLEC12A^{hi} population among cKit⁺CD200R3⁺ basophil-lineage cells showed low expression of CD34 (Fig. 2d and Supplementary Fig S3b), indicating that CLEC12A^{hi} basophils might contain the uni-potent

BaP-like population. Supporting this notion, CD34⁺ BaPs and CD34⁻ CLEC12A^{hi} basophils displayed similar morphology and surface expression profiles (**Supplementary Fig S3c-e**).

(Results: P.7 Line 142-159 in the revised Ms.)

Re-clustering of these three clusters identified 1 cluster of *Fcer1a*⁺*Kit*⁺*Cd34*⁺ pre-BMP-like cells¹², 1 cluster of *Clec12a*^{hi} basophils (**Baso1**) and 2 clusters of *Clec12a*^{lo} basophils (**Baso2 and Baso3**) (**Fig. 2e, f and Supplementary Fig. S4e**). Consistent with the flow cytometric analysis data, *Clec12a*^{hi} basophils (Baso1) were abundant in the bone marrow and much fewer in the spleen (**Fig. 2e, f**). In line with flow cytometric analysis, 21% of *Clec12a*^{hi} Baso1 cells displayed low levels of *Cd34* expression (**Fig. 2f and Supplementary Fig. S4e**). Of note, Seurat clustering of Baso1 cells revealed that *Cd34*⁺ and *Cd34*⁻ populations could not be segregated as distinct clusters (**Supplementary Fig. S5a-c**), suggesting that *Cd34*^{lo} BaP-like cells showed gene expression profiles similar to those of *Cd34*⁻*Clec12a*^{hi} basophils.

To validate our findings, we re-analyzed the publicly available scRNA-seq datasets reported by Weinreb et al.²⁵ (GEO accession number: GSE140802) and identified 1 cluster of *Fcer1a*⁺*Kit*⁺*Cd34*⁺ pre-BMP-like cells, 2 clusters of *Clec12a*^{hi} basophils (Baso1 and Baso2) and 3 clusters of *Clec12a*^{lo} basophils (Baso3, Baso4, and Baso5) (**Supplementary Fig. S6a-b**). Even though 11.4 % of *Clec12a*^{hi} basophils displayed low levels of *Cd34*, Seurat clustering failed to segregate *Cd34*⁺ and *Cd34*⁻ populations as distinct clusters (**Supplementary Fig. S6a-b**) in accordance with the analysis of our scRNA-seq data. Therefore, BaPs do not seem to represent a particular stage of the basophil development in terms of gene profiling.

(Results: P.7 Line 162-164 in the revised Ms.)

Pseudo-time trajectory analysis and RNA velocity analysis of our scRNA-seq data inferred the basophil differentiation trajectory from pre-BMP-like cells to *Clec12a*^{hi} basophils, and then to *Clec12a*^{lo} basophils (**Fig. 2g, h and Supplementary Fig. 7**).

(Results: P.8 Line 189-191 in the revised Ms.)

Accordingly, in the following experiments, we designated CLEC12A^{hi}CD9^{lo} / FcεRIα^{hi}CD49b^{lo} and CLEC12A^{lo}CD9^{hi} / FcεRIα^{lo}CD49b^{hi} basophils as **pre-basophils** and mature basophils, respectively.

(Results: P.9 Line 202-204 in the revised Ms.)

Of note, CD34⁻ pre-basophils displayed a gene expression profile similar to that of CD34⁺ BaPs (**Fig. 4a**), strengthening our forementioned conclusion that CD34⁺ BaPs are included in the CLEC12A^{hi} pre-basophil population in terms of gene profiling.

(Discussion: P.13 Line 302-314 in the revised Ms.)

In the present study, we took advantage of this technology combined with flow cytometric and functional analyses and succeeded in the identification of previously-unappreciated CLEC12A^{hi}CD9^{lo} pre-basophils that are developmentally located downstream of pre-basophil and mast cell progenitors (pre-BMPs) and upstream of CLEC12A^{lo}CD9^{hi} mature basophils in mice. The majority of pre-basophils express little or no expression of CD34 while approximately 10% of them express low levels of CD34 and therefore likely correspond to

BaPs. BaPs were identified in 2005 as unipotent basophil precursors that are downstream of GMPs and differentiate to mature basophils¹⁰. They are characterized by their surface expression of CD34, a transmembrane glycoprotein widely utilized as a marker for hematopoietic progenitor cells³². The scRNA-seq analysis in the present study revealed that the *Cd34* and *Cd34*^{lo} pre-basophils cannot be segregated as distinct clusters. Therefore, we here propose that BaPs do not represent a particular stage of the basophil development and rather constitute a minor fraction of the newly-identified pre-basophil population in terms of gene profiling.

Comment #1:

“In Figure 4 the authors leverage bulk RNAseq to compare BaPs to immature CLEC12A^{hi}CD9^{lo} and mature CLEC12A^{lo}CD9^{hi} basophils. While bulk data does not allow dissection of the heterogeneity within these immature basophils, it can be used to identify transcriptionally similar cells in the authors scRNAseq data. This could be achieved by integrating the bulk data into the scRNAseq data or by calculating scores based on gene lists obtained from the bulk RNAseq data.”

According to the Reviewer’s kind suggestion, we conducted the scoring of the scRNA-seq data (Fig. 2e) by differentially expressing genes between CLEC12A^{hi} pre-basophils and CLEC12A^{lo} mature basophils (Fig. 4a). On one hand, when the mature basophil score was displayed on the feature plot (as shown in Figure a below), the majority of cells in the Baso2 and Baso3 clusters showed the high score, as expected while the small fraction of cells in the Baso1 cluster showed the mature score at low or middle levels. On the other hand, when the pre-basophil score was displayed, the majority of cells in the Baso1 cluster showed the highest score while the substantial fraction of cells in the Baso2 and Baso3 clusters showed the pre-basophil score at various levels. These results suggested that CLEC12A^{hi} pre-basophils contains some mature basophils.

Figure for Comment #1. (a, b) Mature basophil or pre-basophil scores were calculated by using the gene lists significantly upregulated or downregulated in mature basophils compared to pre-basophils in bulk RNA-seq data in Fig. 4a. Feature plots showing the scores for mature and pre-basophils are shown in a and b, respectively.

The cell clustering using the scRNA-seq analysis is based on the expression profile of mRNAs in each cell whereas the cell fractionation using the flow cytometric analysis is based on the expression profile of cell surface markers in each cell. The levels of mRNAs and those of the surface expression of corresponding proteins may not be correlated well with each other, depending on the efficiency of translation, synthesis and degradation of proteins or the stability of proteins on the cell surface. Therefore, we assume that it is practically difficult to fractionate cell populations by flow cytometry in a manner faithfully reflecting the expression profile of their mRNAs.

Fortunately, we were able to identify and segregate CLEC12A^{hi} pre-basophils and CLEC12A^{lo} mature basophils by flow cytometric analysis, based on the differential expression of *Clec12a* among the clusters identified by the scRNA-seq analysis, even though some discrepancy between two types of fractionations was observed as expected.

Comment #2:

“In line with the previous remark, a clear absence of cells in-between the Baso3 and Baso2 populations can be observed in the scRNAseq data obtained from the spleen (Figure 2E). The absence of these cells could be interpreted as them being upstream of the cells detected in both the spleen and bone marrow. Therefore, it would be informative to leverage RNA velocity to provide a prediction with regards to the directionality of development. This should allow the authors to either validate the trajectory predicted by monocle or generate a more accurate trajectory.”

According to the Reviewer’s valuable suggestion, we re-analyzed the scRNA-seq data in Fig. 2e by using scVelo. This analysis predicted the differentiation trajectory in the order of pre-BMP-like cells, *Clec12a*^{hi} Baso2, *Clec12a*^{lo} Baso4 and then Baso3 (Fig. 2g and Supplementary Fig. S7 in the revised manuscript).

Accordingly, we revised Results section as followed.

(Results: P.7 Line 162-164 in revised Ms.)

Pseudo-time trajectory analysis and RNA velocity analysis of our scRNA-seq data inferred the basophil differentiation trajectory from pre-BMP-like cells to *Clec12a*^{hi} basophils, and then to *Clec12a*^{lo} basophils (Fig. 2g, h and Supplementary Fig. S7).

Comment #3:

“These predicted developmental pathways could further be dissected by integrating the scRNAseq data generated in this study with previously published datasets (Dahlin et al, 2018; Weinreb et al 2020). While this might provide insights into the developmental hierarchy of basophil populations described here, it is possible that this approach might not be feasible due to the technical limitations associated with integrating scRNAseq datasets obtained using different technologies.”

According to the Reviewer’ kind suggestion, we sought to integrate our scRNA-seq data (Figure 2e) with the previously published datasets (Dahlin et al *Blood* 2018 and Weinreb et al. *Science* 2020). However, the integration of scRNAseq data seems to be unsuccessful,

because basophils in our scRNA-seq data and those in the publicly available scRNA-seq data were mapped in different clusters (**Fig. a-b** shown below), possibly due to technological limitations.

Importantly, the re-analysis of the scRNA-seq data of mouse total bone marrow cells (Weinreb et al.) identified one cluster of $Cd34^+Kit^+$ pre-BMP-like cells, 2 clusters of $Clec12a^{hi}$ pre-basophils and 3 clusters of $Clec12a^{lo}$ mature basophils (**Supplementary Fig. 6a-c** in the revised manuscript) in accordance with the analysis of our scRNA-seq data. Trajectory analysis further revealed the continuous gene expression changes from pre-BMP-like cells towards pre-basophils and then to mature basophils (**Fig. c** shown below), which validates our results.

Figure for Comment #3. (a) scRNA-seq datasets presented in Fig 2e and scRNA-seq dataset obtained by Dahlin et al. were integrated for further analysis. UMAP plots of scRNA-seq data in Fig2e and Dahlin et al are shown. (b) scRNA-seq datasets presented in Fig 2e and scRNA-seq dataset obtained by Weinreb et al. were integrated for further analysis. UMAP plots of scRNA-seq data in Fig2e and Weinreb et al are shown. (c) Monocle3 pseudotime analysis was conducted on the basophil clusters in scRNA-seq dataset by Weinreb et al. UMAP plot colored by pseudotime is shown.

Accordingly, we added **Supplementary Fig. S6a-c** and revised Results section as follows:

(Results: P.7 Line 152-159 in revised Ms.)

To validate our findings, we re-analyzed the publicly available scRNA-seq datasets reported by Weinreb et al.²⁵ (GEO accession number: GSE140802) and identified 1 cluster of $Fcer1a^+Kit^+Cd34^+$ pre-BMP-like cells, 2 clusters of $Clec12a^{hi}$ basophils (Baso1 and Baso2) and 3 clusters of $Clec12a^{lo}$ basophils (Baso3, Baso4, and Baso5) (**Supplementary Fig. S6a-b**). Even though 11.4 % of $Clec12a^{hi}$ basophils displayed low levels of $Cd34$, Seurat clustering failed to segregate $Cd34^+$ and $Cd34^-$ populations as distinct clusters (**Supplementary Fig. S6a-b**) in accordance with the analysis of our scRNA-seq data. Therefore, BaPs do not seem to represent a particular stage of the basophil development in terms of gene profiling.

Comment #4:

“In Figure S3 a re-analyzed UMAP is visualized that contains cells derived from both the spleen and bone marrow. It would be informative if the authors would be able to visualize $Cd34$ and Kit RNA expression as is the case in Figure S3C. Moreover, cell cycle scores visualized on top of the UMAPs would be informative for the reader as this suggests that certain populations are actively proliferating and potentially developing.”

We thank the Reviewer for bringing this important issue to our attention. According to the Reviewer’s suggestions, we added feature plots indicating the expression of $Cd34$ and Kit

in re-clustered basophil populations in Supplementary Fig. S4e. The expression of *Kit* was mainly observed in pre-BMP-like cells, whereas the expression of *Cd34* was observed in 26% of pre-BMP-like cells and 20% of Baso2 (Supplementary Fig. S4e). Moreover, pre-BMP-like cells and Baso1 displayed high S phase score, as compared with mature basophils in Baso2 and Baso3 clusters (Supplementary Fig. S9).

Accordingly, we added new figures (Supplementary Fig. S4e and Fig. S9b) and revised Results section of manuscript as follows:

(Results: P9. Line 208-209 in revised Ms.)

in accordance with the scRNA-seq analysis showing that *Clec12a*^{hi} Baso1 cluster displayed high S phase score, compared with *Clec12a*^{lo} Baso2/3 clusters (Supplementary Fig. S9).

Comment #5:

“It is unclear as to whether the secretion of IL4 upon stimulation with different cytokines is an inherent function of basophil precursors or can be attributed to residual contaminants within the CLEC12A^{hi}CD9^{lo} population. The authors could leverage ligand-receptor prediction algorithms to predict the receptors and their downstream molecules that are activated by the cytokines leveraged in Figure 4G. These can then be visualized onto the relevant UMAPs to assess potential heterogeneity within the immature basophils. In case these pathways have been thoroughly described, the relevant genes can be directly visualized without the need for ligand-receptor predictions.”

We thank the Reviewer for bringing this important issue to our attention. Since previous study has demonstrated that IL-3-mediated IL-4 production by basophils was dependent on the signals via Fc receptor γ -chain (encoded by *Fcer1g*) and Syk (Hida et al. *Nat Immunol* 2009), we visualized the expression of these signaling molecules as well as the components of IL-3 receptor on UMAP plot as shown in Figure below. Even though the expression of *Il3ra* and *Syk* was higher in *Clec12a*^{hi} pre-basophils, the expression of other genes including *Csf2rb*, *Csf2rb2* and *Fcer1g* was higher in *Clec12a*^{lo} mature basophils. Therefore, we could not detect the clear correlation that can explain the different responsiveness to IL-3 stimulation. Therefore, functional difference between pre-basophils and mature basophils might be attributed to the difference in other unknown signaling components or some post-transcriptional mechanisms.

Figure for Comment #5-1. Feature plots showing the gene expression of IL-3 receptor components and signaling molecules associated with IL-3 signaling are shown.

To resolve the Reviewer’s concern on the contamination of CLEC12A^{lo} mature basophils, we adopted the alternative method. To examine the IL-4 expression of pre-basophils at single cell levels, we utilized IL-4-reporter mice in which IL-4 expression can be tracked by GFP. When stimulated with cytokine stimulation including IL-3, IL-18, IL-33 and LPS, pre-

basophils displayed higher frequency of GFP⁺ cells as compared with mature basophils (as shown in **Figure a-b** below). These data indicate that data in **Fig. 4g** appeared not to be owing to residual contaminant of CLEC12A^{lo} mature basophils.

Figure for Comment #5-2. BMBAAs generated by culturing bone marrow cells of G4 mice in the presence of IL-3 (0.3ng/mL) were stimulated with PBS, IL-3, IL-3+IL-18, IL-3+IL-33, or IL-3+LPS for 6hr and were subsequently subjected to flow cytometric analysis. (a) The expression of CLEC12A and GFP in both CLEC12A^{hi} pre-basophils (colored by blue) and CLEC12A^{lo} mature basophils (colored by red) in stimulated BMBAAs is shown. The frequency of GFP⁺ cells in each basophil subpopulation is also described in this figure. (b) The frequency of GFP⁺ cells among CLEC12A^{hi} pre-basophils and CLEC12A^{lo} mature basophils is shown (mean ± SEM, n = 3 each).

Comment #6:

“In figure 6 the authors describe the role of CXCR4 down regulation that allows immature basophils to emigrate from the bone marrow. The authors however do not leverage their scRNAseq data obtained from bone marrow and spleen to corroborate this. It would be interesting to see CXCR4 expression visualized on the different UMAPs in the case that CXCR4 expression not only changes upon emigration out of the bone marrow but also differs along development.”

According to the Review’s suggestion, we added **Supplementary Fig. S16** that shows the expression of *Cxcr4* in the feature plot of the scRNA-seq data of basophil clusters (Fig.2e).

Accordingly, we revised Results section as follows.

(Results: PP. 11-12 Line 287-290 in the revised Ms.)

The scRNA-seq and flow cytometric analyses of bone marrow cells demonstrated that the expression of CXCR4 was higher in pre-basophils than in mature basophils at both transcript and protein levels (Fig. 6c, **Supplementary Fig. S16**),

Comment #7:

“In both Figure 1D and Figure 5C the UMAPs have not been re-analyzed, limiting the interpretability for the reader. It would be interesting to see these populations in more detail after going through re-analysis.”

According to the Reviewer’s suggestion, we conducted the re-analysis of the scRNA-seq data presented in **Fig. 1d** and **5e** (corresponding to Fig. 5c in the original Ms.). Re-analysis of

basophil clusters in **Fig. 1d** identified one *Clec12a*^{hi} pre-basophil cluster and two *Clec12a*^{lo} mature basophil clusters, as shown in a-b below.

Figure for Comment #7-1. scRNA-seq of BMBAs presented in Fig.1d were re-analyzed. UMAP plot of BMBAs is shown in a. Feature plots showing the expression of indicated genes are shown in b.

Re-analysis of basophil clusters in **Fig. 5e** identified one *Clec12a*^{hi} pre-basophil cluster and two *Clec12a*^{lo} mature basophil clusters, as shown in a-c below.

Figure for Comment #7-2. scRNA-seq dataset of basophils from second Nb-infected mice presented in Fig.5e was re-analyzed. UMAP plots of basophils from the bone marrow and skin infection site were separately shown in a. The frequency of each cluster is shown in b. Feature plots showing the expression of indicated genes in combined dataset of the bone marrow and skin are shown in c.

Comment #8:

“In figure 5C it is unclear whether the CD9^{lo}CLEC12A^{hi} cells derived from the skin are enriched to equal numbers. Clarifying this would facilitate the interpretation of the figure. In addition, it is unclear whether the labels immature and mature are computationally defined (eg clusters) or if these correspond to the sorting strategy used.”

Pre-basophils, Mature1 and Mature2 labeled in Fig. 5e are computationally-defined Seurat clusters (not fractionated by cell sorting). The frequency of each cluster is now shown as bar graphs in the revised Fig. 5e (corresponding to Fig. 5c in the original Ms.).

Accordingly, we revised Figure Legends section as follows.

(Figure Legends: P. 35 Line 759-761 in the revised Ms.)

In e, UMAP plots show 3 basophil clusters (pre-basophil, Mature 1 and 2; **defined by Seurat clustering**) in the left panel. **The frequency of each cluster is shown in the right panel.**

Comment #9:

“It is unclear whether cKit-CD200R+CD9^{hi}CLEC12A+CD34+ BpAs were excluded during sort purification prior to the adoptive transfer experiment in Figure 3C. Stating this in the methods, figure or legend would facilitate the interpretation of that figure.”

According to the Reviewer’s suggestion, we added that information to Methods section as follows.

(Methods: P.19 Line 477-478 in the revised Ms.)

In this experiment, CD34⁺ cells were not excluded from CLEC12A^{hi} basophils.

REVIEWER COMMENTS

Reviewer #1 (Remarks to the Author):

The revision has overall improved the manuscript. However, a new major concern arose as consequence of the revision. As now written, the manuscript makes a bold statement that tries to downplay the existence of the previously defined CD34+ BaPs as a particular stage of basophil development, both in the abstract and throughout the manuscript. This is an unnecessary statement, as no adequate proof is provided. Instead, it leads to an intricate line of reasoning, contradicting published studies (including one that was published in Nature Communications during the author revision process), and it also leads to several ambiguous statements and conclusions. The issues are presented here:

1. Gating of CD34+ BaP as in Fig S3a results in a 94.7 % CLEC12A+ cell population. By contrast, CD34- Ba are only 36.8 % CLEC12A+. Thus, the study itself clearly shows that the gating of CD34+ BaP is indeed a good way to capture pure CLEC12A+ cells – a population that the study presents as a cell stage. The flow cytometry analysis in Fig S3d-e hints that the CD34+ BaP are the most immature “pre-basophils” (even though the reviewer is aware that the results are not conclusive, likely as the development is a continuum).

2. The authors' response to “the Reviewer's main concern” describes that the Cd34- and Cd34lo basophils expressing CLEC12Ahi cannot be segregated as individual clusters. This reasoning is based on the results presented in Fig S6 (data from Weinreb et al). However, for this reasoning to be true, one needs to accept that cells annotated as “pre-BMP-like” (indicated as cells expressing Cd34, Kit, and Fcεr1a on the gene expression level) are correctly annotated and disregarded from in the analysis, and one also needs to assume that the protein levels and gene expression levels of all cells match to 100 %. The authors' response to comment 1 makes the same assumptions, quoted “[the] CD34+ BaPs” express Kit in addition to Cd34, therefore most likely correspond to Kit+CD34+ pre-BMPs...”. Yet, no evidence is presented to support the assumption that Fcεr1a+ Kit+ Cd34+ cells in Hamey et al and Weinreb et al constitute pre-BMPs (instead of BaP). Notably, low levels of c-Kit are even found on CD34+ BaP in the present manuscript (Fig S3d-e). The meaning of the results presented in Fig S5 is also unclear, as Cd34 expression is polarized to the upper part of cluster 0 (Basophil1) in Fig S4, suggesting that Cd34 helps segregating the entire basophil population. Furthermore, Matsumura et al (Nature Communications, 2022, DOI: <https://doi.org/10.1038/s41467-022-34906-1>) recently demonstrated that Kit and Cd34 gene expression is continuously downregulated throughout basophil development, and expression is detected until the last stages of basophil differentiation in mouse bone marrow. Together, these observations indicate that the assumption in the present manuscript and in the rebuttal is inaccurate. In general, caution is warranted when interpreting data based on gene expression when populations are defined using flow cytometry and surface markers.

The reviewer agrees with the interpretation that the CLEC12A levels can be used to separate early and late stages of basophil development, and the correlation between the CLEC12A marker expression is not 100 % compared with the CD34 marker expression (though there is clear overlap, as discussed above). Additional proof needs to be provided to state that “BaP [...] do not represent a particular stage of the basophil development”, as Matsumura et al, Hamey et al, and Arinobu et al (PNAS, <https://doi.org/10.1073/pnas.0509148102>) all support the observation that CD34 defines the early stages of basophil development at the cell fate, protein and/or gene expression levels.

To conclude: The reviewer can see two options:

A) New experimental evidence needs to be presented to prove that the previously described CD34+ BaPs do not represent cells of a particular stage of basophil differentiation (as described above).
B) The story is revised to clearly highlight the overlap with the CD34+ BaP, but also the non-overlap (CLEC12Ahi CD34neg). This includes changing the text and figures, as discussed above. With the data presented, the reviewer would likely interpret the results as follows: CLEC12A is present on CD34+ BaP, and CLEC12A is then downregulated after the disappearance of CD34 (before becoming mature Ba, see Fig S3d).

Minor comments: Matsumura et al (Nature Communications, 2022, <https://doi.org/10.1038/s41467-022-34906-1>) should be cited.

Reviewer #2 (Remarks to the Author):

The authors have done an excellent job addressing my concerns and the manuscript is acceptable for publication.

Reviewer #3 (Remarks to the Author):

I am supportive of the authors work being published in Nature Communications.

Point-by-point response to Reviewer1:

“The revision has overall improved the manuscript. However, a new major concern arose as consequence of the revision.”

We greatly appreciate the reviewer’s comments and valuable suggestions.

Responding to the Reviewer’s main concern

“As now written, the manuscript makes a bold statement that tries to downplay the existence of the previously defined CD34+ BaPs as a particular stage of basophil development, both in the abstract and throughout the manuscript. This is an unnecessary statement, as no adequate proof is provided.”

We thank the Reviewer for bringing this important issue to our attention. We agree that the statement “BaPs do not represent a particular stage of the basophil development” might be an overstatement, possibly misleading readers. Therefore, we deleted that sentence and modified Abstract, Results and Discussion section in the revised manuscript as follows:

(Abstract: P.4 Line 23-25 in the revised Ms.)

The transcriptomic analysis **predicted** that the pre-basophil population includes previously-defined basophil progenitor (BaP)-like cells in terms of gene expression profile. ~~Basophil progenitors constitute a minor fraction of pre-basophils and do not represent a particular stage of the basophil development.~~

(Results: P.7 Line 151-154 in the revised Ms.)

A small fraction (11.4 %) of *Clec12a*^{hi} basophils displayed low levels of *Cd34* in accordance with the analysis of our scRNA-seq data, **suggesting this small population within *Clec12a*^{hi} basophils may correspond to *Cd34*^{lo} BaP-like cells.** ~~Therefore, BaPs do not seem to represent a particular stage of the basophil development in terms of gene profiling.~~

(Discussion: P.13 Line 302-303 in the revised Ms.)

The transcriptomic analysis predicted that the pre-basophil population include previously-defined basophil-progenitor (BaP)-like cells in terms of gene expression profile. ~~The majority of pre-basophils express little or no expression of CD34 while approximately 10% of them express low levels of CD34 and therefore likely correspond to BaPs. BaPs were identified in 2005 as unipotent basophil precursors that are downstream of GMPs and differentiate to mature basophils¹⁰. They are characterized by their surface expression of CD34, a transmembrane glycoprotein widely utilized as a marker for hematopoietic progenitor cells³³. The scRNA-seq analysis in the present study revealed that the *Cd34* and *Cd34*^{lo} pre-basophils cannot be segregated as distinct clusters. Therefore, we here propose that BaPs do not represent a particular stage of the basophil development and rather constitute a minor fraction of the newly-identified pre-basophil population in terms of gene profiling.~~

(Discussion: P.14 Line 341-343 in the revised Ms.)

In conclusion, we identified previously-unappreciated, c-Kit-CLEC12A^{hi} pre-basophils that are located downstream of pre-BMPs and upstream of mature basophils in the basophil ontogeny. ~~The transcriptomic analysis revealed that previously-defined BaPs constitute a minor fraction of pre-basophils and do not represent a particular stage of the basophil development.~~

*“The authors’ response to “the Reviewer’s main concern” describes that the *Cd34*- and *Cd34*^{lo} basophils expressing CLEC12A^{hi} cannot be segregated as individual clusters. This reasoning is based on the results presented in Fig S6 (data from Weinreb et al). However, for this reasoning to be true, one needs to accept that cells annotated as “pre-BMP-like”*

(indicated as cells expressing Cd34, Kit, and Fcεr1a on the gene expression level) are correctly annotated and disregarded from in the analysis, and one also needs to assume that the protein levels and gene expression levels of all cells match to 100 %...Together, these observations indicate that the assumption in the present manuscript and in the rebuttal is inaccurate. In general, caution is warranted when interpreting data based on gene expression when populations are defined using flow cytometry and surface markers.”

We greatly appreciate the Reviewer’s valuable comments and suggestions. The Reviewer appointed out in the original manuscript “the gene expression profiles of BaP and immature basophils (now designated pre-basophils) appear strikingly similar”. Accordingly, the Reviewer asked us “Further evidence that immature basophils represent a unique population (separate from BaP) is therefore warranted”. This was the Reviewer’s main concern. In the revised manuscript, we showed a possible reason for this similarity: the transcriptomic analysis predicted that the pre-basophil population includes BaP-like cells in terms of the gene expression profile.

As the Reviewer suggested, the statement “the *Cd34*⁺ and *Cd34*^{lo} pre-basophils cannot be segregated as distinct clusters” might be overstatement, as the prediction is based on Seurat clustering under a certain analytical condition and the assumption that levels of gene expression and surface expression of corresponding proteins correlate each other. Therefore, we removed that statement together with Fig. S5 and modified Results and Discussion section in the revised manuscript as follows:

(Results: P.7 Line144-147 in revised Ms.)

In line with flow cytometric analysis, 21% of *Clec12a*^{hi} Baso1 cells displayed low levels of *Cd34* expression (Fig. 2f and Supplementary Fig. S4f). ~~Of note, Seurat clustering of Baso1 cells revealed that *Cd34*⁺ and *Cd34*^{lo} populations could not be segregated as distinct clusters (Supplementary Fig. S5a-c), suggesting that *Cd34*^{lo} BaP-like cells were included in *Clec12a*^{hi} Baso1 cells and showed gene expression profiles similar to those of *Cd34*⁺ *Clec12a*^{hi} basophils.~~

(Results: P.7 Line 151-154 in the revised Ms.)

A small fraction (11.4 %) of *Clec12a*^{hi} basophils displayed low levels of *Cd34*, ~~Seurat clustering failed to segregate *Cd34*⁺ and *Cd34*^{lo} populations as distinct clusters (Supplementary Fig. S6a-e)~~ in accordance with the analysis of our scRNA-seq data, ~~suggesting this small population within *Clec12a*^{hi} basophils may correspond to *Cd34*^{lo} BaP-like cells.~~

(Discussion: P.13 Line 302-303 in the revised Ms.)

~~The transcriptomic analysis predicted that the pre-basophil population include previously-defined basophil-progenitor (BaP)-like cells in terms of gene expression profile. The majority of pre-basophils express little or no expression of CD34 while approximately 10% of them express low levels of CD34 and therefore likely correspond to BaPs. BaPs were identified in 2005 as unipotent basophil precursors that are downstream of GMPs and differentiate to mature basophils¹⁰. They are characterized by their surface expression of CD34, a transmembrane glycoprotein widely utilized as a marker for hematopoietic progenitor cells³³. The scRNA-seq analysis in the present study revealed that the *Cd34*⁺ and *Cd34*^{lo} pre-basophils cannot be segregated as distinct clusters. Therefore, we here propose that BaPs do not represent a particular stage of the basophil development and rather constitute a minor fraction of the newly-identified pre-basophil population in terms of gene profiling.~~

Response to Minor Comment.

“Matsumura et al (Nature Communications, 2022, <https://doi.org/10.1038/s41467-022-34906-1>) should be cited.”

We cited Matsumura et al, in Introduction section of revised Ms as follows:

(Introduction: P.3 Line 58-59 in revised Ms.)

This was also the case in basophils, and dynamic gene expression changes from HSCs to basophils have been reported in mice and humans ¹⁸⁻²².

(References: P. 22 Line 565-566 in revised Ms.)

22. Matsumura, T. et al. A Myb enhancer-guided analysis of basophil and mast cell differentiation. *Nat Commun* 13, 7064 (2022).

REVIEWERS' COMMENTS

Reviewer #1 (Remarks to the Author):

The authors have adequately addressed the reviewer's concerns.